# The chromatin insulator CTCF regulates HPV18 transcript splicing and differentiation-dependent late gene expression

Jack Ferguson[1⊙], Karen Campos-León[1⊙], Ieisha Pentland[1], Joanne D. Stockton[2], Thomas Günther[3], Andrew D. Beggs[1,2], Adam Grundhoff[3], Sally Roberts[1], Boris Noyvert[1,4‡], Joanna L. Parish[1‡*]

1 Institute of Cancer and Genomic Sciences, College of Medical and Dental Sciences, University of Birmingham, Birmingham, West Midlands, United Kingdom, 2 Genomics Birmingham, University of Birmingham, Birmingham, West Midlands, United Kingdom, 3 Heinrich-Pette Institute, Leibniz Institute for Experimental Virology, Hamburg, Germany, 4 CRUK Birmingham Centre and Centre for Computational Biology, University of Birmingham, Birmingham, West Midlands, United Kingdom

⊙ These authors contributed equally to this work.
‡ These authors are joint senior authors on this work.
* j.l.parish@bham.ac.uk

## Abstract

The ubiquitous host protein, CCCTC-binding factor (CTCF), is an essential regulator of cellular transcription and functions to maintain epigenetic boundaries, stabilise chromatin loops and regulate splicing of alternative exons. We have previously demonstrated that CTCF binds to the E2 open reading frame (ORF) of human papillomavirus (HPV) 18 and functions to repress viral oncogene expression in undifferentiated keratinocytes by co-ordinating an epigenetically repressed chromatin loop within HPV episomes. Keratinocyte differentiation disrupts CTCF-dependent chromatin looping of HPV18 episomes promoting induction of enhanced viral oncogene expression. To further characterise CTCF function in HPV transcription control we utilised direct, long-read Nanopore RNA-sequencing which provides information on the structure and abundance of full-length transcripts. Nanopore analysis of primary human keratinocytes containing HPV18 episomes before and after synchronous differentiation allowed quantification of viral transcript species, including the identification of low abundance novel transcripts. Comparison of transcripts produced in wild type HPV18 genome-containing cells to those identified in CTCF-binding deficient genome-containing cells identifies CTCF as a key regulator of differentiation-dependent late promoter activation, required for efficient E1^E4 and L1 protein expression. Furthermore, our data show that CTCF binding at the E2 ORF promotes usage of the downstream weak splice donor (SD) sites SD3165 and SD3284, to the dominant E4 splice acceptor site at nucleotide 3434. These findings demonstrate that in the HPV life cycle both early and late virus transcription programmes are facilitated by recruitment of CTCF to the E2 ORF.

**Data Availability Statement:** All relevant data are within the manuscript and its Supporting Information files.

**Funding:** This work was funded by grants from the Medical Research Council awarded to JLP and SR (MR/R022011/1, MR/T015985/1 and MR/N023498/1). BN is funded through the Cancer Research UK Birmingham Centre award C17422/A25154. The funders had no role in study design, data collection and interpretation, or the decision to submit the work for publication.

**Competing interests:** The authors have declared that no competing interests exist.

## Author summary

Oncogenic human papillomavirus (HPV) infection is the cause of a subset of epithelial cancers of the uterine cervix, other anogenital areas and the oropharynx. HPV infection is established in the basal cells of epithelia where a restricted programme of viral gene expression is required for replication and maintenance of the viral episome. Completion of the HPV life cycle is dependent on the maturation (differentiation) of infected cells which induces enhanced viral gene expression and induction of capsid production. We previously reported that the host cell transcriptional regulator, CTCF, is hijacked by HPV to control viral gene expression. In this study, we use long-read mRNA sequencing to quantitatively map the variety and abundance of HPV transcripts produced in early and late stages of the HPV life cycle and to dissect the function of CTCF in controlling HPV gene expression and transcript processing.

## Introduction

Human papillomaviruses (HPVs) are a family of small, double-stranded DNA viruses that infect cutaneous and mucosal epithelia. Most HPV types cause benign epithelial hyperproliferation, which is usually resolved by host immune activation. However, persistent infection with a subset of HPV types (e.g., HPV16 and 18) is the cause of epithelial tumours including cervical and other anogenital cancers, and carcinoma of the oropharyngeal tract [1].

The viral genome is maintained and replicated in the cell nucleus as an extrachromosomal, chromatinised episome which allows the epigenetic regulation of viral transcription in an equivalent manner to host genes [2]. The regulation of HPV gene expression in differentiating epithelia is tightly regulated and is a key strategy in the maintenance of persistent infection. Several distinct transcriptional start sites (TSSs) have been identified including the major early and late promoters, the E8 promoter ($P_{E8}$) and less well-defined TSSs around nucleotide 520 ($P_{520}$) and 3000 ($P_{3000}$). The relative activity of these promoters is dependent on the differentiation status of the host keratinocyte [3–5]. Establishment of HPV infection occurs in the undifferentiated basal keratinocytes of epithelia where viral genome copy number and transcription are maintained at low levels, presumably to prevent host immune activation. We and others have shown that the viral episome is maintained in an epigenetically repressed state in undifferentiated keratinocytes, characterised by low abundance of trimethylation of lysine 4 (H3K4Me3) and enrichment of trimethylation of lysine 27 (H3K27Me3) on histone H3, which attenuate viral gene expression [5, 6]. The host cell chromatin-organising and transcriptional insulation factor, CCCTC-binding factor (CTCF) is important in the maintenance of the epigenetic repression of the HPV genome through the stabilisation of a chromatin loop. CTCF binds to a conserved site in the E2 open reading frame (ORF) of HPV18 approximately 3,000 base pairs downstream of the viral transcriptional enhancer situated in the long control region (LCR) [7]. Although the major CTCF binding site and the viral enhancer are physically separated, we demonstrated that abrogation of CTCF binding resulted in inappropriate epigenetic activation of the HPV18 enhancer and early promoter (termed $P_{102}$ in HPV18) and increased expression of the viral oncoproteins E6 and E7 (E6/E7) [6, 7]. CTCF physically associates with the transcriptional repressor Yin Yang 1 (YY1) [8] and we subsequently showed that CTCF-dependent epigenetic repression of the HPV18 episome was through interaction with YY1 bound at the viral LCR, such that CTCF and YY1 co-operate to stabilise an epigenetically repressed chromatin loop within the early gene region [6]. While the association of CTCF with the HPV18 episome is not significantly altered by keratinocyte differentiation, YY1 protein

expression and binding to the HPV18 genome is dramatically reduced in differentiated keratinocytes leading to loss of CTCF-YY1 dependent chromatin loop stabilisation, although no differentiation-dependent changes in CTCF protein expression were observed [6]. This differentiation-dependent topological change in the HPV episome is coincident with epigenetic activation of the $P_{102}$ promoter and increased expression of the HPV E6/E7 oncoproteins. Interestingly, HPV18 E7 protein has also been shown to physically associate with YY1. It is unclear whether this contributes to (de)regulation of HPV transcription but an E7-YY1 complex was shown to positively regulate expression of the host gene *lnc-FANCI-2* which may have important implications in HPV-mediated carcinogenesis [9].

Activation of the major late promoter (termed $P_{811}$ in HPV18) in part occurs through epigenetic derepression of the HPV episome upon keratinocyte differentiation [5, 6, 10] and reviewed in [11]. This restricts expression of the viral capsid proteins L1 and L2 to the upper compartment of infected epithelia, limiting their potential for host immune activation [4, 12, 13]. The late promoter also regulates expression of viral intermediate genes including E1, E2, E1^E4 and E5, which are important for viral genome amplification in the upper layers of the infected epithelia [14, 15]. The mechanisms underlying the differentiation-dependent epigenetic activation of late promoter activity are not clear, but it has been shown that the viral enhancer in the LCR is required for late promoter activation [16] and that differentiation-dependent enhancement of transcription elongation may play a key role in late promoter activation [17].

Further enhancing the complexity of HPV gene expression regulation, the polycistronic HPV mRNA is subject to extensive post-transcriptional splicing, which gives rise to an array of transcripts that each encode a distinct subset of full length, and/or fusion proteins. While studies have mapped the HPV18 transcriptome [18–20], the quantification of HPV promoter activity and the abundance of each mature transcript has not been reported. Cellular splicing factors are utilised and manipulated by the virus to co-ordinate differentiation dependent viral transcript splicing, including the serine-arginine rich (SR) proteins and heterogeneous ribonucleoproteins (hnRNPs) [20–22]. In addition to its functions in chromatin looping and epigenetic isolation, CTCF can play an important role in regulating alternative gene splicing, most likely through multiple mechanisms. In the host cell *CD45* locus, CTCF binding within exon 5 promotes inclusion of upstream exons by creating a "roadblock" to pause RNA polymerase II progression, allowing more efficient recognition of weak exons by the splicing machinery [23]. It has also been shown that DNA methylation-dependent binding of CTCF within normally weak exons promotes inclusion during co-transcriptional splicing [24]. To support these findings, a significant enrichment of CTCF binding sites in close proximity to alternatively spliced exons has been reported [25]. However, CTCF binding at distant sites can also influence alternative exon usage through the stabilisation of intragenic chromatin loops [26]. Our early analysis of CTCF-dependent control of HPV18 transcript splicing indicated an important role for this factor in maintaining the complexity of splicing events [7] but the global effect of CTCF on HPV18 transcript processing was not analysed.

Next generation sequencing (NGS) has revolutionised virology research by providing nucleotide resolution data on existing and emerging pathogens, prevalence, and evolution. However, conventional Illumina-based RNA sequencing (RNA-Seq) methods are limited in that information on the structure of full-length transcripts, including alternative splicing is sacrificed to preserve accuracy and read depth [27]. Direct, long-read Nanopore sequencing overcomes this limitation by providing quantitative data on the abundance of individual mRNA isoforms [28].

In this study, we use Nanopore sequencing to quantify the spectrum of HPV18 transcripts in HPV18 episome-containing primary human keratinocytes and to map differentiation-

induced changes in promoter usage, splicing and transcript abundance. Furthermore, we characterise the global effect of CTCF binding to the HPV18 genome on transcript splicing and early and late promoter activity.

## Methods

### Ethical approval

The collection of neonatal foreskin tissue for the isolation of primary human foreskin keratinocytes (HFKs) for investigation of HPV biology was approved by Southampton and South West Hampshire Research Ethics Committee A (REC reference number 06/Q1702/45). Written consent was obtained from the parent/guardian. The study was approved by the University of Birmingham Ethical Review Committee (ERN 16–0540).

### Cell culture, methylcellulose differentiation and organotypic raft culture

Normal primary HFKs from neonatal foreskin epithelia were transfected with recircularised HPV18 wild type (WT) or -ΔCTCF genomes and maintained on irradiated J2-3T3 fibroblasts in complete E medium [29] as previously described [7]. For methylcellulose-induced keratinocyte differentiation, 3 x$10^6$ HPV18 or ΔCTCF-HPV18 genome containing keratinocytes were suspended in E-media supplemented with 10% FBS and 1.5% methylcellulose and incubated at 37˚C, 5% $CO_2$ for 48 hrs. Cells were then harvested by centrifugation at 250 x g followed by washing with ice-cold PBS. Cells were then either suspended in medium containing 1% formaldehyde to cross-link for chromatin immunoprecipitation (ChIP) as described below, or DNA, RNA and protein was extracted from cell pellets as previously described [7]. Southern blotting was carried out as previously described [6].

Organotypic raft cultures were prepared as previously described [7]. Rafts were cultured for 14 days in E medium without epidermal growth factor to allow cellular stratification. Raft cultures were fixed in 3.7% formaldehyde and paraffin embedded and sectioned by Propath Ltd (Hereford, United Kingdom).

### Antibodies

Anti-CTCF (61311) and anti-H4Ac (39925) antibodies was purchased from Active Motif and used at 5–8 μg/sample for ChIP alongside mouse anti-FLAG (M2; Sigma Aldrich) as a negative control. For immunofluorescence staining, HPV18 L1 (5A3) antibody was purchased from Nova Costra (used at 1:100) and rabbit polyclonal E1^E4 antisera (1:5000), were produced as previously described [30]. Alexa-488 and –594 conjugated anti-rabbit/mouse secondary antibodies (Invitrogen) were used at 1:1000. For Western blotting, anti-GAPDH (6C5; 1:5000) was purchased from Santa Cruz. HPV18-specific antibodies were as follows: mouse E1^E4 (1D11; 1:10 [30]), E6 was purchased from Santa Cruz (G$^{-7}$; 1:50), E7 was purchased from Abcam (8E2; 1:100) and sheep anti-E2 antisera (1:1000) were produced as previously described [31]. Involucrin antibody (SY5) was purchased from Sigma Aldrich and used at 1:1000. HRP-conjugated anti-mouse and anti-rabbit secondary antibodies (Jackson Laboratories) were used 1:5000.

### Chromatin immunoprecipitation-qPCR (ChIP-qPCR)

ChIP-qPCR assays were performed using the ChIP-IT Express Kit (Active Motif) as per the manufacturer's protocol. Briefly, cells were fixed in 1% formaldehyde for 5 mins at room temperature with gentle rocking, quenched in 0.25 M glycine and washed with ice-cold PBS. Nuclei were released using a Dounce homogeniser. Chromatin shearing was carried out by

**Table 1. Primer sequences used for ChIP-qPCR experiments. Ta, annealing temperature; bp, base pairs.**

| Primer pair (amplicon mid-point) | Amplicon length (bp) | Forward (5'– 3') | Reverse (5'– 3') | Ta (˚C) |
|---|---|---|---|---|
| 4539 | 198 | GGGGTCGTACAGGGTACATT | GATGTTATATCAAACCCAGACGTG | 56 |
| 5479 | 196 | TCTGCCTCTTCCTATAGTAATGTAACG | GGAATAAAATAATATAATGGCCACAAA | 56 |
| 5753 | 195 | CCTCCTTCTGTGGCAAGAGT | GGTCAGGTAACTGCACCCTAA | 56 |
| 6746 | 175 | AGTCTCCTGTACCTGGGCAA | AACACCAAAGTTCCAATCCTCT | 58 |
| 7363 | 123 | GTGTGTTATGTGGTTGCGCC | GGATGCTGTAAGGTGTGCAG | 58 |
| 7796 | 99 | ACTTTCATGTCCAACATTCTGTCT | ATGTGCTGCCCAACCTATTT | 56 |
| 224 | 140 | TGTGCACGGAACTGAACACT | CAGCATGCGGTATACTGTCTC | 58 |
| 819 | 136 | CGAACCACAACGTCACACAAT | ACGGACACACAAAGGACAGG | 58 |
| 1418 | 70 | GCAATGTATGTAGTGGCGGC | TACACTGCTGTTGTTGCCCT | 58 |
| 2884 | 131 | TGCAGACACCGAAGGAAACC | CATTTTCCCAACGTATTAGTTGCC | 58 |
| 3022 | 191 | GGCAACTAATACGTTGGGAAAA | TGTCTTGCAGTGTCCAATCC | 56 |
| 3221 | 113 | AGGTGGCCAAACAGTACAAGT | GCCGTTTTGTCCCATGTTCC | 58 |
| 3478 | 194 | TGGGAAGTACATTTTGGGAATAA | TCCACAGTGTCCAGGTCGT | 56 |
| 4029 | 102 | TATGTGTGCTGCCATGTCCC | CTGTGGCAGGGGACGTTATT | 56 |

sonication at 25% amplitude for 30 secs on/30 secs off for a total time of 15 mins using a Sonics Vibracell sonicator fitted with a microprobe. ChIP efficiency was assessed by qPCR using SensiMix SYBR master mix using a Stratagene Mx3005P (Agilent Technologies, Santa Clara, CA, USA). Primer sequences for ChIP experiments are shown in Table 1. Cycle threshold ($C_T$) values were used to calculate fold enrichment compared to a negative control FLAG antibody with the following formula:

$$\text{Fold binding over IgG} = (2^{\Delta C_T \text{ Target}})/(2^{\Delta C_T \text{ IgG}})$$

Where $\Delta C_T$ target = Input $C_T$−Target $C_T$ and $\Delta C_T$ IgG = Input $C_T$−IgG $C_T$. Each independent experiment was performed in technical triplicate and data shown are the mean and standard deviation of three independent repetitions.

## ChIP-Seq

ChIP and respective input samples were used for generation of ChIP-Seq libraries as described [32]. Briefly, 2–10 ng DNA was used in conjunction with the NEXTflex Illumina ChIP-Seq library prep kit (Cat# 5143–02) as per the manufacturer's protocol. Samples were sequenced on a HiSeq 2500 system (Illumina) using single read (1x50) flow cells. Sequencing data was aligned to the HPV18 genome (accession number: AY262282.1) using Bowtie [33] with standard settings and the -m1 option set to exclude multi mapping reads [34].

*Alignment to human genome*: Similar to HPV, CTCF and input ChIP-seq reads of two independent infections with either HPV18 or ΔCTCF-HPV18 were aligned to the human reference genome hg19 using Bowtie. Reads mapping to multiple host loci were excluded. CTCF peaks were called using MACS1.4 for the individual replicates using input material as background control. Peaks were stringently filtered and kept only if present in the two replicate samples of either wild type or mutant. Overlapping CTCF peak regions between wild type infection and infection with the mutant virus were detected by bedtools. Quantification, scatter plots for correlation analysis and visualization were performed in EaSeq (https://easeq.net/).

## RNA sequencing and data analysis

For RNA-Seq, libraries were prepared using Tru-Seq Stranded mRNA Library Prep kit for NeoPrep (Illumina, San Diego, CA, USA) using 100ng total RNA input according to manufacturer's instructions. Libraries were pooled and run as 75-cycle–pair end reads on a NextSeq 550 (Illumina) using a high-output flow cell. Sequencing reads were aligned to human (GRCh37) and HPV18 (AY262282.1) genomes with STAR aligner (v2.5.2b) [35]. The computations were performed on the CaStLeS infrastructure [36] at the University of Birmingham. Sashimi plots were generated in Integrative Genomics Viewer (IGV), Broad Institute (http://software.broadinstitute.org/software/igv/).

## Nanopore direct RNA sequencing and data analysis

$8x10^7$ cells from undifferentiated or methylcellulose differentiated keratinocytes containing HPV18 (WT or ΔCTCF) samples for RNA extraction using the RNeasy Plus Mini Kit (Qiagen) according to the manufacturer's instructions and DNaseI treated (Promega). 500 ng of polyA + RNA was used in conjunction with the direct RNA sequencing kit (Oxford Nanopore technologies, Oxford, UK [SQK-RNA002]). All protocol steps are as described in [37]. The reads were aligned to the human (GRCh37) and HPV18 (AY262282.1) genomes using minimap2 [38] with options "-ax splice -uf -k14" for nanopore direct RNA mapping. The splicing coordinates were extracted from the bam files using custom scripts. HPV18 transcripts were included in the dataset when a minimum threshold of three reads per million in at least two samples was achieved to ensure that each transcript was identified at least four times in multiple samples. Illumina and Nanopore data sets used in this study are available at the European Nucleotide Archive (http://www.ebi.ac.uk/ena/data/view/PRJEB47821).

## Quantitative RT-PCR

cDNA was synthesised using Superscript III (Invitrogen) according to the manufacturer's instructions. qPCR was performed using a Stratagene Mx3005P detection system with SyBr Green incorporation and the primers listed in Table 2.

## Cell lysis and western blotting

Cells were lysed with urea lysis buffer (ULB; 8 M urea, 100 mM Tris-HCl, pH 7.4, 14 mM ß-mercaptoethanol, protease inhibitors) and protein concentration determined. Protein extracts from organotypic raft cultures were harvested using ULB and homogenised using a Dounce homogeniser contained with a category II biological safety cabinet. Lysates were incubated on ice for 20 mins before centrifugation at 16,000 x g for 20 mins at 4°C. Supernatant was transferred to a fresh tube and protein concentration assessed by Bradford Assay. For Western blotting, equal quantities of protein lysates were separated by SDS-PAGE and western blotting was carried out by conventional methods. Chemiluminescent detection was carried out using a Fusion FX Pro and densitometry performed with Fusion FX software.

**Table 2. Primer sequences used for qRT-PCR experiments. Ta, annealing temperature; bp, base pairs.**

| Primer set name | Amplicon length (bp) | Forward (5'– 3') | Reverse (5'– 3') | Ta (°C) |
|---|---|---|---|---|
| 3165^3434 | 129 | CTGCTTTAAAAAAGTACCAGTGA | GCCGACGTCTGGCCGTAGGTCTTTGCGG | 60 |
| 3284^3434 | 129 | CATGGGACAAAACTACCAGTGACG | GCCGACGTCTGGCCGTAGGTCTTTGCGG | 60 |
| E1^E4 | 126 | GATCCAGAAATACCAGTGACG | GCCGACGTCTGGCCGTAGGTCTTTGCGG | 60 |

## Immunofluorescence

Immunofluorescence was carried out on paraffin embedded organotypic raft culture sections using the agitated low temperature epitope retrieval (ALTER) method as previously described [39]. Briefly, slides were sequentially immersed in Histoclear (Scientific Laboratory Supplies) and 100% IMS and incubated at 65˚C in 1 mM EDTA (pH 8.0), 0.1% Tween 20 overnight with agitation. Slides were then blocked in PBS containing 20% heat-inactivated normal goat serum and 0.1% BSA (Merck). Primary antibodies were diluted in block solution and incubated over-night at 4˚C followed by 3x PBS washes. Fluorophore-conjugated secondary antibodies were diluted in block buffer and added to slides which were incubated at 37˚C for 1 hour. Slides were subsequently washed 4x 10 mins in PBS with Hoechst 33342 solution (10 μg/ml) added to the final PBS wash. Slides were mounted in Fluoroshield (Sigma-Aldrich) and visualised using a Nikon inverted Epifluorescent microscope fitted with a 40x oil objective. Images were captured using a Leica DC200 camera and software.

## Results

We have previously characterised a CTCF binding site within the E2 open reading frame (ORF) of HPV18 which is strongly bound by CTCF in a primary HFK model of the HPV18 life cycle (**Fig 1A**) [6, 7]. Although the E2-CTCF binding site was the most CTCF enriched region of the HPV18 genome in our ChIP-qPCR analysis, there did appear to be other regions of the viral genome that were bound at a lower level by CTCF. In addition, CTCF binding sites have been predicted in the late gene region of HPV18 and other high-risk HPV types and binding has been demonstrated in HPV31 episomes [7, 40]. To analyse CTCF binding to the HPV18 genome with greater sensitivity, we opted to map CTCF binding peaks using ChIP-sequencing (ChIP-Seq). Anti-CTCF immunoprecipitated chromatin harvested from HFKs harbouring HPV18 episomes was subject to Illumina next generation sequencing. Reads were aligned to the HPV18 genome revealing robust enrichment of CTCF in the E2 ORF with maxi-mal binding between nucleotides 2960–3020, corresponding to the previously identified E2-CTCF binding site (**Fig 1B**). No other distinct CTCF peaks were observed in the HPV18 genome. In addition, ChIP-Seq analysis of CTCF enrichment in ΔCTCF-HPV18 genomes in which the E2-CTCF binding site was mutated to prevent CTCF binding by the introduction of three conservative nucleotide substitutions that did not alter the E2 protein sequence (**Fig 1A**; herein termed ΔCTCF-HPV18), revealed a complete loss of CTCF binding to the E2-ORF with no evidence of enhanced binding at secondary sites (**Fig 1B**), confirming our previous ChIP-qPCR analysis of this mutant virus. These findings were consistent in two independent HFK donors.

Having established that ΔCTCF-HPV18 episomes do not bind CTCF at the E2-ORF or any other secondary site(s), we sought to determine whether CTCF recruitment to HPV18 epi-somes altered the distribution of binding sites within the host genome. This was achieved by comparison of CTCF binding peaks within the cellular genome of HPV18 HFKs to ΔCTCF-HPV18 HFKs in two independent keratinocyte donors. The total number of CTCF binding peaks identified were 36,808 and 36,378 for HPV18 and ΔCTCF-HPV18, respectively (**S1A Fig**) and this was consistent in an independent keratinocyte donor. Heatmap analysis of all CTCF peaks demonstrated no obvious difference in the distribution of CTCF binding in HPV18 compared the ΔCTCF-HPV18 (**S1B and S1C Fig**). These data provide evidence that sequestration of CTCF protein to HPV18 episomes *per se* does not affect CTCF function in the regulation of host cell gene expression.

Our previous studies showed that abrogation of CTCF binding at the HPV18 E2 ORF resulted in increased transcriptional activity of the HPV18 early promoter ($P_{102}$) and a

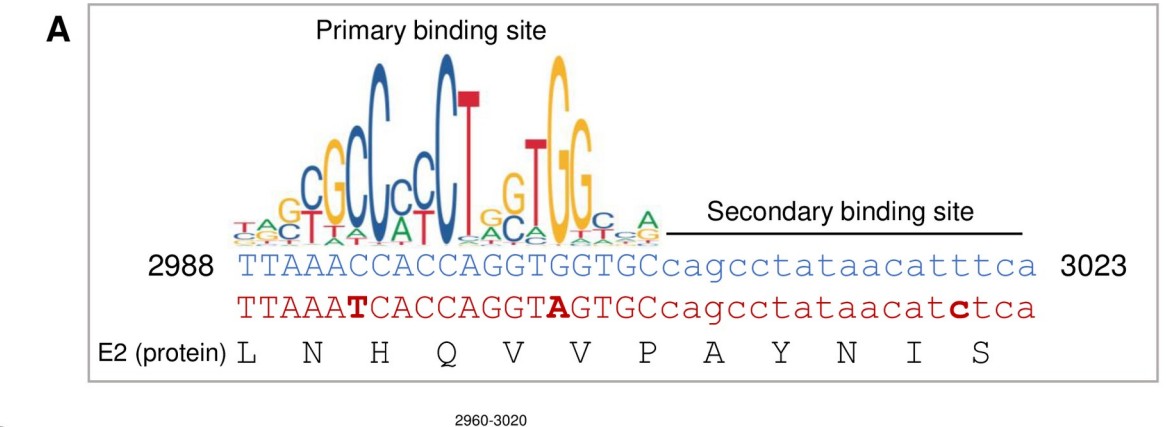

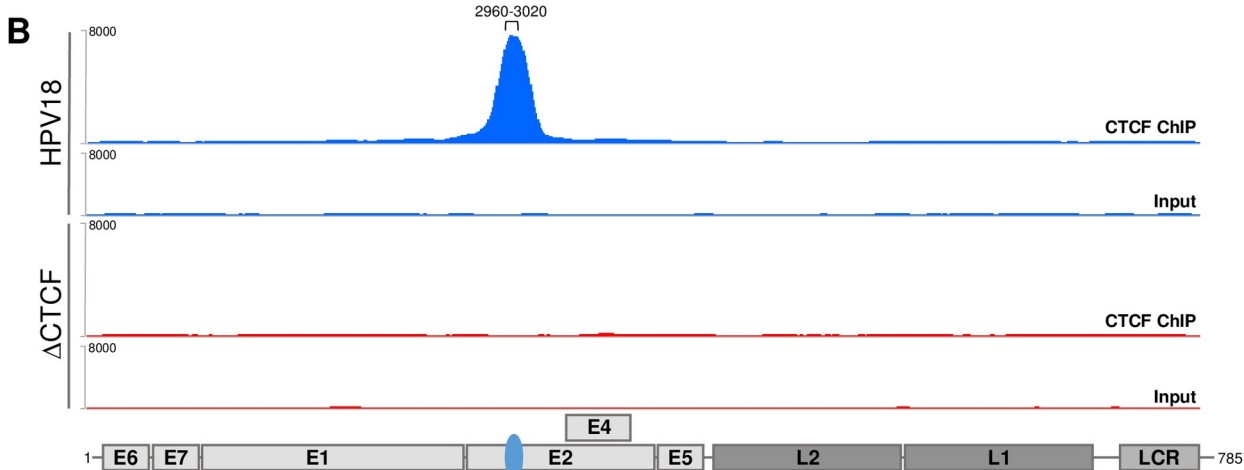

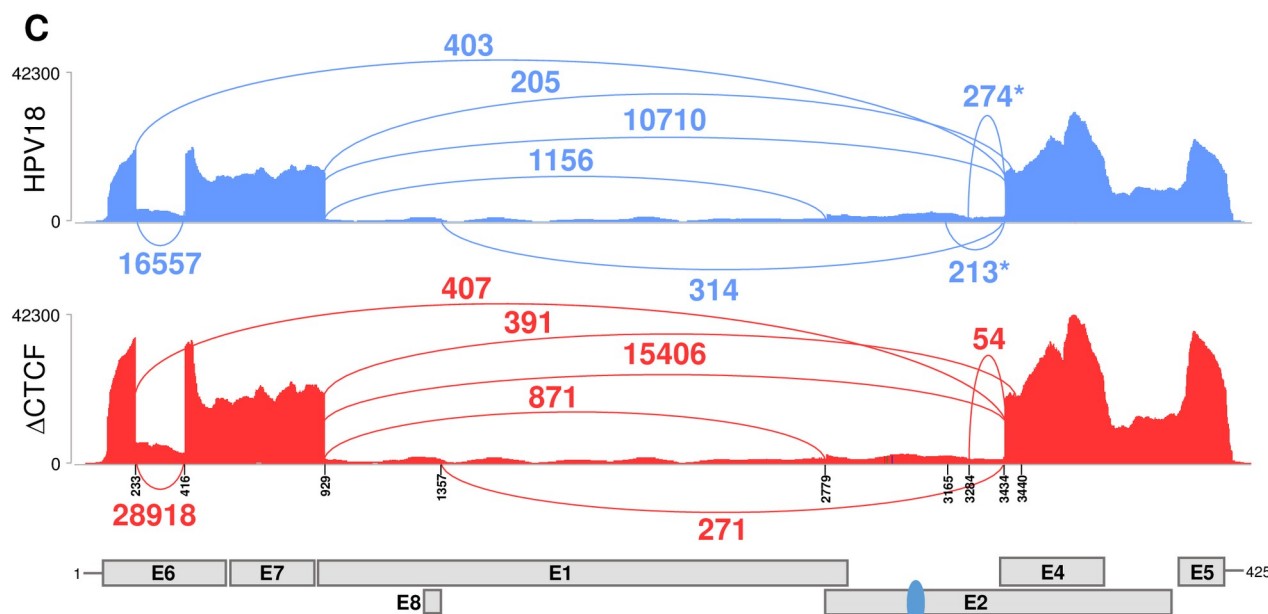

**Fig 1. Abrogation of CTCF recruitment to the HPV18 E2 ORF alters early transcript splicing.** (A) Nucleotide sequence of the CTCF binding site identified in the E2 ORF (nucleotides 2988–3023; blue text). The primary and secondary CTCF binding sites are shown as detailed in [46]. Conservative nucleotide substitutions introduced in ΔCTCF-HPV18 mutant (red text) are shown in bold. The E2 protein sequence (black text) is

unaltered. Graphical representation of the primary CTCF binding site motif was obtained from JASPER2018 (http://jaspar.genereg.net/). (B) Enrichment of CTCF in the HPV18 genome was assessed by ChIP-Seq in either HPV18 (blue) or ΔCTCF-HPV18 (red) genome-containing keratinocytes. Next generation sequencing data were visualised using IGV. The position of HPV18 ORFs, LCR and E2-CTCF binding site (blue oval) are indicated below the alignment profiles. (C) Exon-exon junctions in Illumina RNA-Seq data sets of either HPV18 (blue) or ΔCTCF-HPV18 (red) genome-containing keratinocytes were identified and quantified in IGV and represented in Sashimi plots. The co-ordinates of splice donor and acceptor sites and annotated ORFs are indicated. The number of reads at each exon-exon junction is indicated. *denotes splicing event identified in HPV18 but reduced or not detected in ΔCTCF-HPV18 genome containing cells.

concomitant increase in E6/E7 protein expression [6, 7]. These studies also revealed alterations in the splicing of early transcripts, indicated by a significant reduction in the abundance of transcripts spliced at 233^3434 upon amplification by semi-quantitative RT-PCR [7]. To confirm these findings and to further characterise CTCF-dependent regulation of HPV18 transcript splicing, we utilised high-depth Illumina RNA-Seq data in HPV18 and ΔCTCF-HPV18 transfected primary HFKs to quantify individual splicing events (**Fig 1C**). While there were a similar number of splicing events at 233^3434 in the HPV18 and ΔCTCF-HPV18 genome-containing cells (403 and 407 events, respectively), splicing at 233^416 was increased in ΔCTCF-HPV18 genome containing cells in comparison to wild type (28,918 events compared to 16,557 events respectively, Fisher's test p-value <0.00001), which could account for the observed relative reduction in amplification of transcripts spliced at 233^3434 by qRT-PCR [7]. Interestingly, we also noted a reduction in splicing at 3284^3434, previously proposed to encode a truncated form of the E2 protein, E2C and a complete loss of splicing at 3165^3434 in ΔCTCF-HPV18 genome containing cells compared to wild type HPV18. Found at relatively low abundance, splicing at 3165^3434 has been previously described and predicted to encode a novel E2^E4 fusion protein termed E2^E4L [41]. Similarly, splicing at 2853^3434 has been proposed to encode a shorter form of E2^E4 fusion protein, E2^E4S [41], however, this splice was not detected in our Illumina RNA-Seq data. These findings suggest that CTCF may play a role in controlling acceptor site usage downstream of the E2-CTCF binding site.

While individual splicing events can be quantified using conventional short-read RNA sequencing methods, the evaluation of the structure of individual transcripts and the multiple splicing events that occur within a single transcript is not possible. To fully characterise and, for the first time, quantify the relative abundance of individual HPV18 transcripts in primary HFKs, purified and polyA+ enriched RNA was analysed by direct long-read Nanopore sequencing. Cells were either grown in monolayer culture on feeder cells (undifferentiated) or embedded in semi-solid methylcellulose containing medium for 48 hours, to induce synchronous differentiation.

Previous analysis has demonstrated that ΔCTCF-HPV18 episomes are maintained at similar copy number to wild type HPV18 in undifferentiated keratinocytes [6, 7]. Differentiation of keratinocytes induces amplification of HPV18 episomes, which was confirmed by Southern blotting in both HPV18 and ΔCTCF-HPV18 genome-containing HFKs (**Fig 2A**) and this was consistent in an independent keratinocyte donor (**S2 Fig**). To ensure induction of cellular markers of differentiation, host transcripts were quantified and normalised as reads per million (RPM) for each sample. Principal component analysis (PCA) showed very little variance in host cell gene expression between HPV18 and ΔCTCF-HPV18 before and after differentiation, but clear separation in principal component 1 upon differentiation of both cell populations (**S3A Fig**). Induction of a cellular marker of keratinocyte differentiation, involucrin (IVL) was observed in HPV18 (**Fig 2B**; Fisher's test p-value < 0.00001) and ΔCTCF-HPV18 HFKs (**S3B Fig**). In addition, an alteration in expression and transcript splicing of the keratinocyte-specific extracellular matrix protein, ECM1, upon keratinocyte differentiation has been reported [42]. Undifferentiated keratinocytes express full length ECM1 transcript 2 but

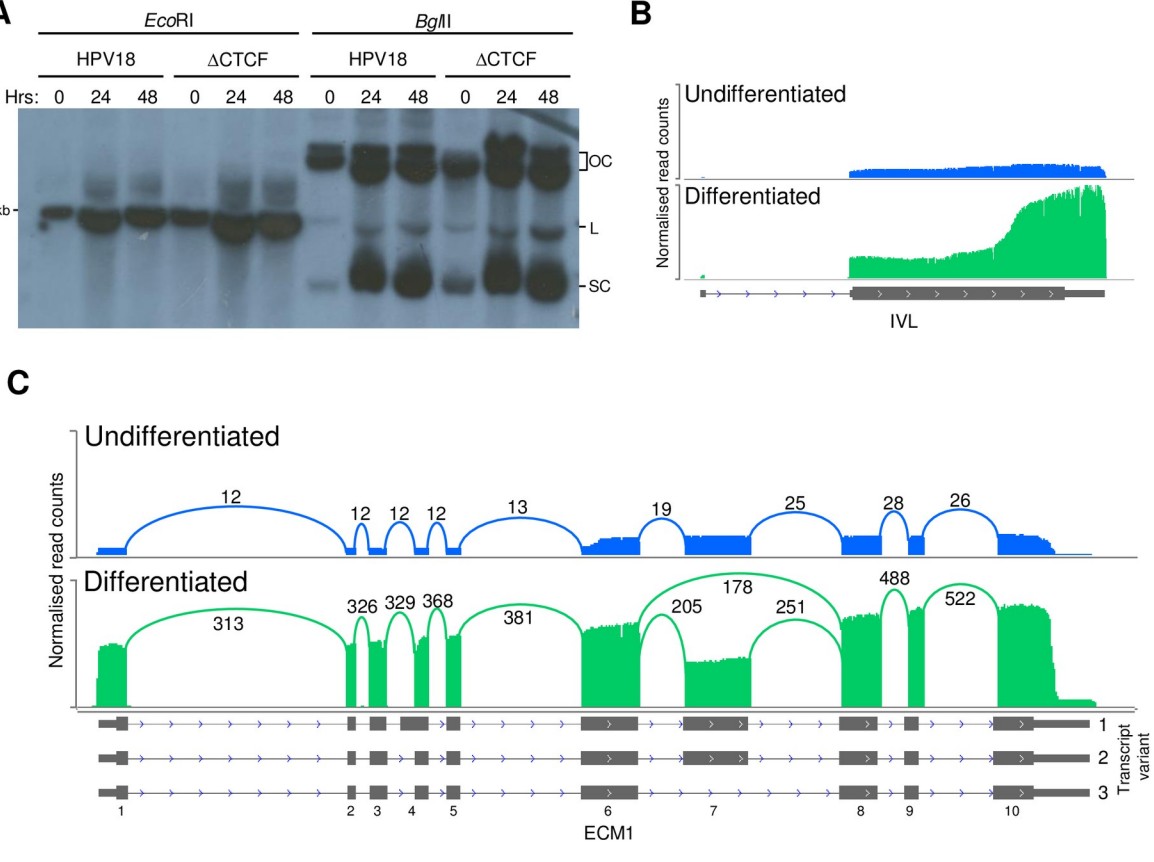

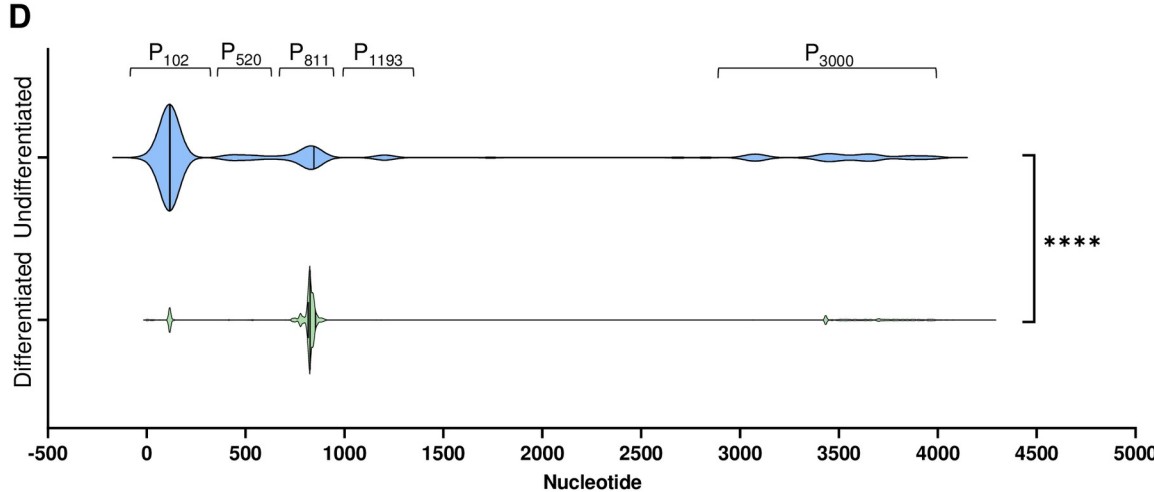

**Fig 2. Analysis of differentiation-dependent host cell gene expression and HPV transcriptional start site usage.** HPV18-HFK were synchronously differentiated in methylcellulose for 48 hrs. (A) Amplification of HPV18 and ΔCTCF-HPV18 episomes was detected by Southern blotting following digestion with *Eco*RI to linearise the HPV18 episomes, or *Bgl*II which digests cellular DNA only (OC, open circle; L, linear; SC, supercoiled). (B-D) Host and viral transcriptomes in undifferentiated (blue) and differentiated (green) HPV18-HFK were analysed by long read Nanopore RNA-Seq, demonstrating enhanced involucrin (IVL) expression following keratinocyte differentiation (B) and enhanced ECM1 expression combined with differentiation-induced exon 7 skipping in transcript variant 3; exon numbering and transcript variants are indicated to the right and below the ECM1 gene annotation. (D) Clustered HPV18 promoter usage in undifferentiated and differentiated keratinocytes showing differentiation-dependent alteration of the major early ($P_{102}$) and major late ($P_{811}$) promoter usage. ****$p < 0.0001$ (Fisher's test).

expression of a shorter, alternatively spliced transcript (transcript 3) is induced upon keratinocyte differentiation. Analysis of ECM1 transcripts in our Nanopore sequencing data demonstrated the appearance of ECM1 transcript 3 which lacks exon 7 in methylcellulose differentiated keratinocytes only (**Figs 2C and S3C**). Furthermore, gene set enrichment analysis of host cell gene expression changes induced by synchronous differentiation of both HPV18 and ΔCTCF-HPV18 genome-containing cells revealed a significant enrichment of biological processes including keratinocyte differentiation and epithelial cell differentiation (**S3D Fig**), with broadly consistent alteration of genes involved in keratinocyte differentiation in both HPV18 and ΔCTCF-HPV18 HFKs (**S4 Fig**).

Virus host fusion transcripts were identified at very low abundance (<2% of total HPV reads), indicative of low-level viral integration, with no obvious differences in the spectrum of integration sites identified in HPV18 or ΔCTCF-HPV18 HFKs (**S1 and S2 Tables**, and **S5 Fig**). Nonetheless, these fusion transcripts were removed from our data set prior to analysis to include only those transcripts derived from HPV episomes. Data were then normalised to the total number of reads in each sample to calculate RPM of each viral transcript species. In agreement with previous reports [18, 19], five clear groupings of transcriptional start regions were identified in undifferentiated HPV18 genome containing cells, which originated between nucleotides 1–350 ($P_{102}$), 351–700 ($P_{520}$), 701–900 ($P_{811}$), 1000–1400 ($P_{1193}$) and 2800–4000 ($P_{3000}$) (**Fig 2D**) at previously described transcriptional promoters [18, 19], which were used to define transcript species in subsequent quantifications. Keratinocyte differentiation resulted in a significant change in promoter usage characterised by activation of the $P_{811}$ major late promoter (**Fig 2D**). In undifferentiated HPV18 genome-containing cells, the most abundant transcript was initiated at the $P_{102}$ promoter and spliced at 233^416–929^3434 (transcript 3; **Fig 3**). This transcript has the potential to encode E6*I, E7, E1^E4 and E5. Several novel transcripts were identified above our inclusion threshold of at least three individual reads in at least two samples including transcripts 10 and 22, which have the potential to encode E6*I, E7 and E5. Although these transcripts have not been previously described, the specific splicing combination only includes previously annotated splice sites, but in a previously undetected combination. As they are low abundance, these transcripts are unlikely to be of major biological significance. Interestingly, splicing at both 3165^3434 and 3284^3434 was observed in undifferentiated and differentiated HPV18 cells (transcripts 8 and 9; **Fig 3**). However, these transcripts originated from the $P_{3000}$ promoter and therefore lack the E2 start codon at nt2816 and more likely encode E5 in the basal keratinocytes rather than E2^E4 fusion proteins as previously suggested [41].

Comparison of viral transcripts in HPV18 and HPV18-ΔCTCF genome-containing cells revealed a significant increase in abundance of the major early transcript 3, which encodes E6*I, E7, E1^E4 and E5 (**Fig 3**, Fisher's test p-value < 0.00001). A more modest increase in the second most abundant transcript in undifferentiated cells, originating from the $P_{102}$ promoter and spliced at 929^3434 was also observed, which has the potential to encode full length E6 as well as E7, E1^E4 and E5 (transcript 4; **Fig 3**, Fisher's test, non-significant). The increased abundance of these major early viral transcripts corroborates the previously observed increase in E6 and E7 protein expression when CTCF binding site is ablated [6, 7]. Transcripts spliced at 929^3440 (transcripts 10, 11 and 12) were also detected at low abundance. Notably, splicing at both 3165^3434 and 3284^3434 (transcripts 8 and 9; **Fig 3**) was significantly reduced in undifferentiated and differentiated HPV18-ΔCTCF genome containing cells compared to HPV18 (Fisher's test p-value < 0.00001 and 0.01, respectively) corroborating our finding in Illumina RNA-Seq datasets that CTCF may function to enhance the activity of downstream weak SD sites in the HPV18 genome. The reduction in splicing at 3165^3434 and 3284^3434 was validated by qRT-PCR using primers specific to these splice events. A significant reduction

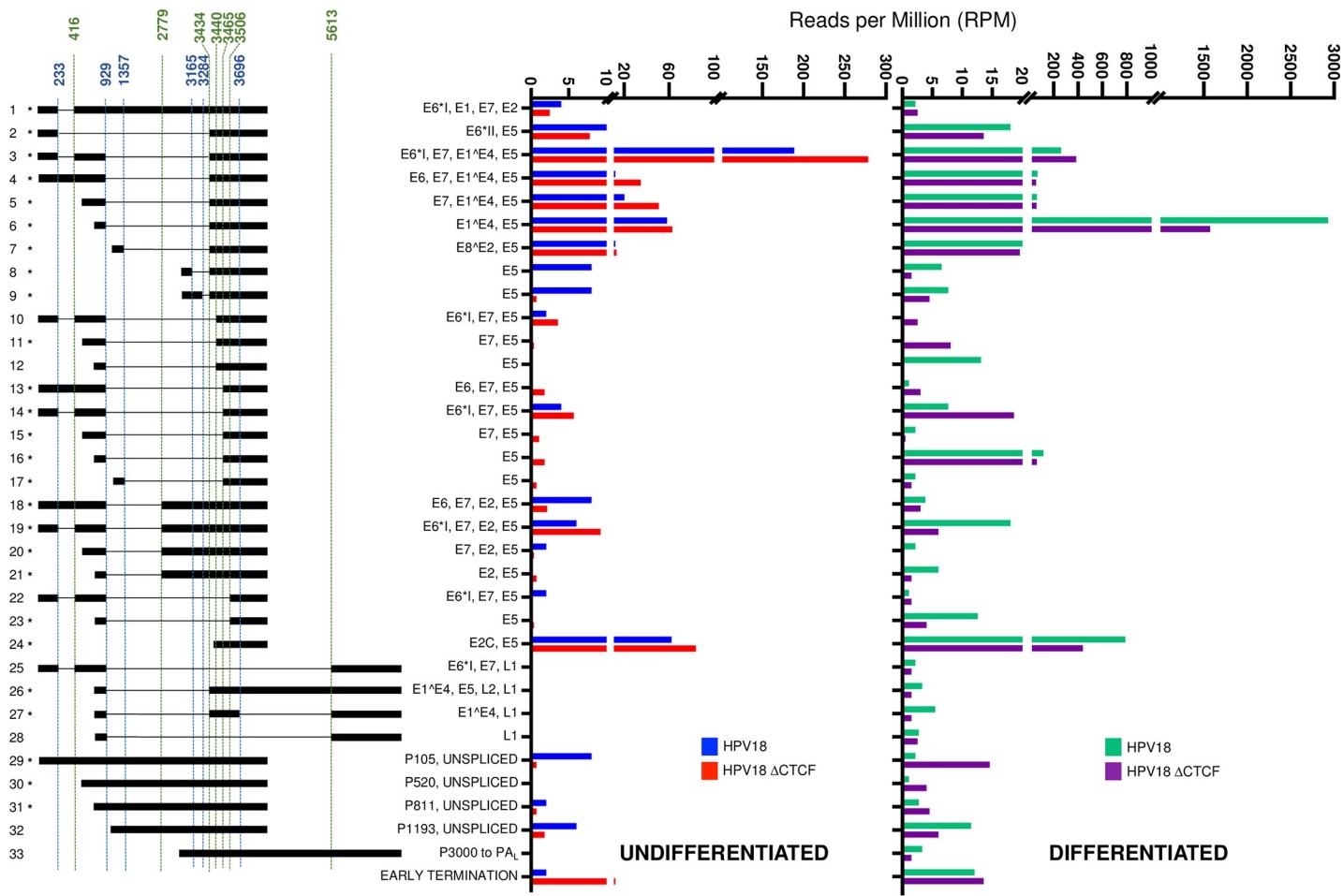

**Fig 3. Quantitative analysis of the HPV18 transcriptome in undifferentiated and differentiated keratinocytes and alterations induced by abrogation of CTCF binding.** Alignment of Nanopore direct RNA sequencing data to the HPV18 genome facilitated the characterisation of all HPV-specific transcripts. Transcripts were included in the data set if they were represented by three or more individual reads in at least two samples. The relative abundance of each transcript type was calculated in reads per million (RPM) of the total reads in each sample. Relative abundance (RPM) of each transcript is shown for HPV18 (blue) and ΔCTCF-HPV18 (red) genome-containing cells in undifferentiated keratinocytes (left) and for HPV18 (green) and ΔCTCF-HPV18 (purple) in differentiated keratinocytes (right). Splice donor (blue) and acceptor (green) sites are indicated above the transcript map and HPV18 ORFs encoded by each transcript are shown. *denotes transcripts that have previously been identified [18, 19].

in 3165^3434 spliced transcripts was observed in undifferentiated and differentiated ΔCTCF-HPV18 episome containing cells in comparison to wild type and this was consistent in two independent HFK donors (**Fig 4A**). Similarly, splicing at 3284^3434 was reduced in ΔCTCF-HPV18 episomes. Although this reduction did not reach significance in undifferentiated HFK donor 1, the reduction was significant in donor 2 and in both donors following differentiation (**Fig 4B**). Together, these data show that abrogation of CTCF binding within the E2 ORF of HPV18 results in reduced splicing between the downstream weak splice donor sites SD3165 and SD3284 and the dominant spice acceptor site SA3434.

Transcripts that originate from the $P_{811}$ late promoter were abundantly expressed in undifferentiated cells; transcripts originating from this promoter and spliced at 929^3434 to encode E1^E4 and E5 proteins (transcript 6; **Fig 3**) were the second most abundant transcript in undifferentiated cells. As expected, the abundance of this transcript was dramatically increased around 50-fold (Fisher's test p-value < 0.00001) upon differentiation of HPV18 cells in

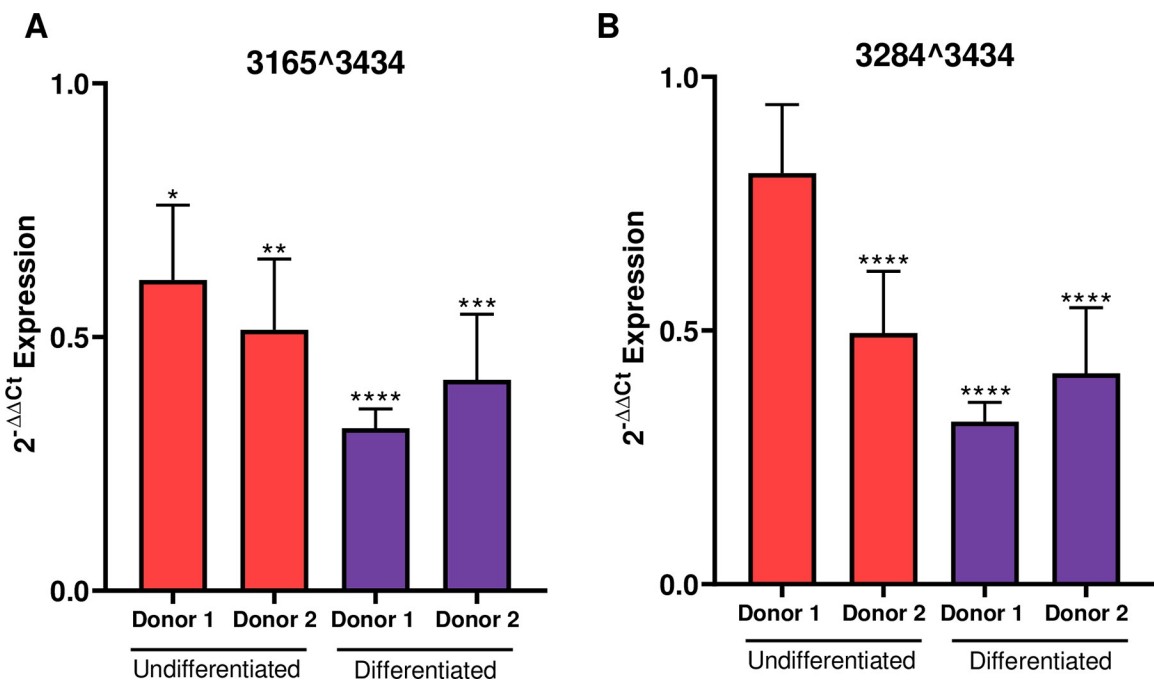

**Fig 4. Loss of CTCF binding at the E2-ORF of HPV18 causes reduced downstream transcript splicing.** Splicing at (A) 3165^3434 and (B) 3284^3434 was assessed by qRT-PCR in two independent keratinocyte donors cultured in monolayer (undifferentiated; red) and in methylcellulose for 48 hrs (differentiated; purple). Data shown are the mean and standard error of transcript abundance normalised to β-actin and expressed and fold expression ($2^{-\Delta\Delta CT}$) compared to donor matched HPV18 episome containing cells.

methylcellulose. However, while differentiation of HPV18-ΔCTCF genome-containing cells similarly resulted in an increase in abundance of this major E1^E4 encoding transcript, the overall abundance of this transcript was reduced by around 50% compared to HPV18. It is also interesting to note that transcripts encoding the L1/L2 capsid proteins (transcripts 25–28; **Fig 3**) were induced upon cellular differentiation in HPV18 genome-containing cells, albeit at a low level, but these transcripts were all lower in abundance in HPV18-ΔCTCF cells. These data suggest that recruitment of CTCF to the HPV18 genome at the E2-ORF may be important for differentiation-dependent activation of the viral late promoter.

The major transcriptional promoters in the HPV18 genome have been previously mapped using 5' RACE [18]. Although transcript sequencing by Nanopore does not provide nucleotide resolution accuracy in mapping transcription start sites [43], the clustering of the 5' end of viral transcripts was clearly enriched at the previously annotated viral promoters (**Fig 2D**). Therefore, to characterise the differential activity of the major viral promoters in HPV18 and ΔCTCF-HPV18 cells, the 5' end of each viral read in our Nanopore datasets was mapped and quantified. The 5' end of most transcripts (>90%) mapped in the region of three previously described promoters; $P_{102}$, $P_{811}$ and $P_{3000}$ (**Fig 5**). Interestingly, the 5' end of transcripts that originated from both the $P_{102}$ and $P_{811}$ promoters clustered as a sharp peak at the previously annotated transcriptional start site whereas the 5' end of transcripts originating from the $P_{3000}$ promoter were more broadly distributed (**Fig 5A, 5B and 5C**). As expected, the $P_{102}$ promoter was the most active promoter in undifferentiated HPV18 genome-containing cells with very few transcripts originating from the $P_{811}$ late promoter. Differentiation of these cells resulted in a dramatic increase in transcripts originating from the $P_{811}$ promoter (Fisher's test p-value < 0.00001), coincident with a slight increase in $P_{102}$ activity (Fisher's test p-value < 0.00001) (**Fig 5A and 5B**). Transcripts originating from the $P_{102}$ promoter were ~30%

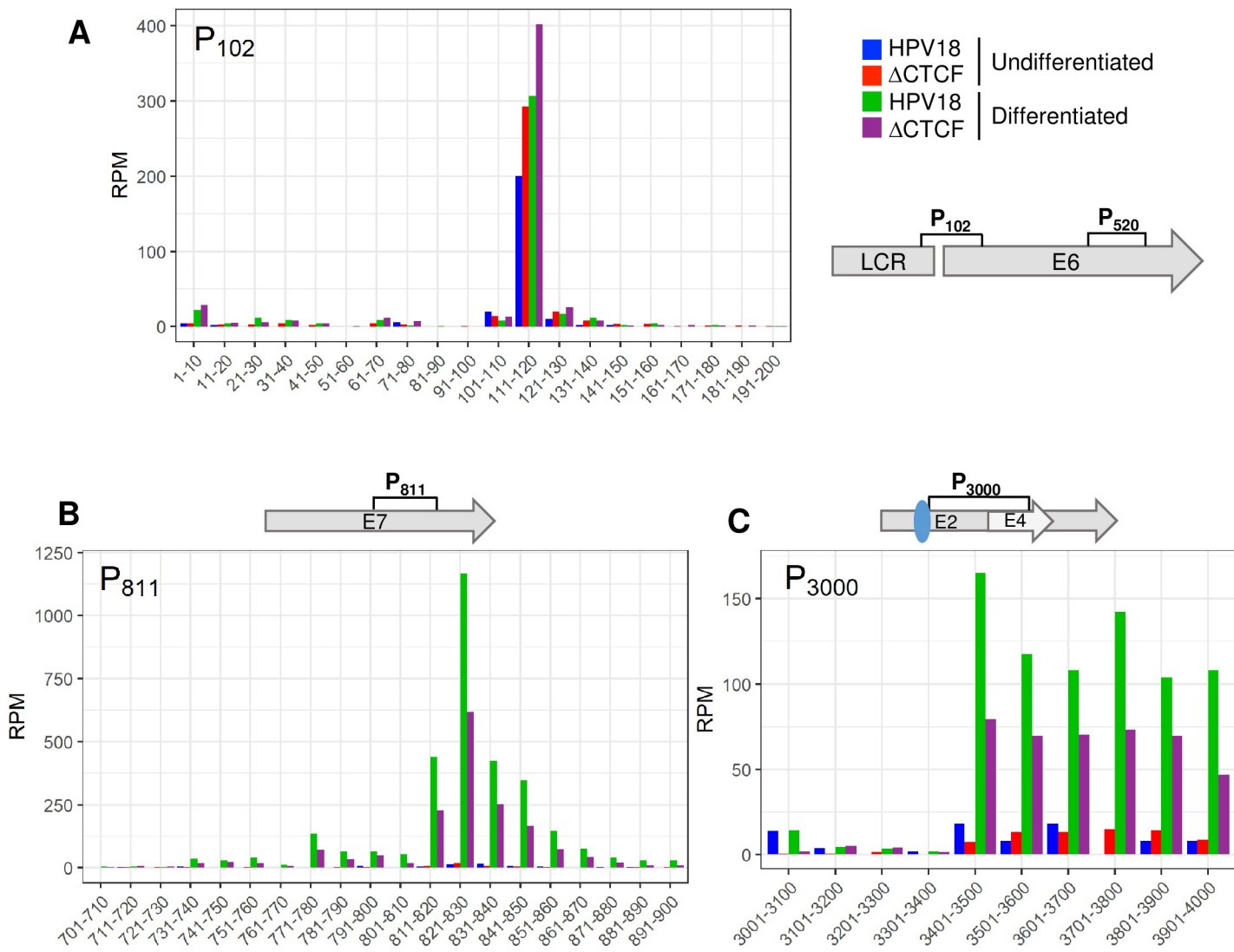

**Fig 5. Quantitative analysis of transcription start site usage in undifferentiated and differentiated keratinocytes and CTCF-dependent regulation of promoter activity.** The 5' end of each HPV18 transcript was identified in Nanopore RNA sequencing data sets and relative abundance calculated as reads per million (RPM). Total counts at each nucleotide position were binned into 10 (A and B) or 100 (D) nucleotide regions in the data shown. Transcripts originating around the $P_{102}$ (A), $P_{811}$ (B) and $P_{3000}$ (C) promoters were identified in HPV18 and ΔCTCF-HPV18 cells in undifferentiated (blue and red, respectively) and methylcellulose differentiated (green and purple, respectively) cultures. Relevant HPV18 genome features are shown alongside each panel. The E2-CTCF binding site is indicated by a blue oval.

more abundant in HPV18-ΔCTCF genome containing cells than HPV18, which was further activated upon cellular differentiation confirming enhanced activity of the early promoter in the absence of CTCF recruitment. Interestingly, the activity of the $P_{811}$ late promoter was notably lower in differentiated ΔCTCF-HPV18 genome containing cells compared to HPV18 (Fisher's test p-value < 0.00001), providing evidence that the activity of the late promoter in differentiated cells is attenuated when CTCF recruitment is abrogated. Very few transcripts originated from $P_{3000}$ in undifferentiated cells, however this promoter was strongly activated following cellular differentiation in HPV18 genome containing cells. As was observed at $P_{811}$, differentiation-dependent activation of $P_{3000}$ was reduced in ΔCTCF-HPV18 genome containing cells compared to HPV18. The $P_{E8}$ ($P_{1193}$) and $P_{520}$ promoters were only weakly active with less than 10% of transcripts originating at these promoters in undifferentiated cells and

the activity of these promoters was not altered by keratinocyte differentiation or mutation of the E2-CTCF binding site.

Analysis of promoter usage in the bulk population of viral transcripts revealed that while there was a greater proportion of transcripts which initiated from the $P_{102}$ early promoter in ΔCTCF-HPV18 episomes than HPV18 (indicated by tighter density grouping and increased slope of the violin plot kernel), this did not reach significance (p = 0.16) (**Fig 6A**). In contrast, highly significant differences were observed between promoter usage in ΔCTCF-HPV18 episomes compared to HPV18 following keratinocyte differentiation (p < 1E-16). While in HPV18 cells, the promoter usage density was highly enriched at the $P_{811}$ promoter, transcripts in ΔCTCF-HPV18 genome-containing cells were less abundant at the $P_{811}$ promoter, and the $P_{102}$ promoter was proportionately more active than in HPV18 episomes (**Fig 6B**). These analyses demonstrate that differentiation-dependent stimulation of $P_{811}$ major late promoter activity is facilitated by recruitment of CTCF to the E2 ORF.

To determine whether the reduced differentiation-dependent activation of $P_{811}$ in ΔCTCF-HPV18 genomes resulted in reduced late protein expression, we analysed E1^E4 transcript and protein abundance in methylcellulose differentiated cultures. The reduction in E1^E4 transcript abundance in ΔCTCF-HPV18 in comparison to HPV18 following differentiation was validated in two independent keratinocyte donors by qRT-PCR (**Fig 7A**). Western blotting of lysates harvested from HPV18 and ΔCTCF-HPV18 genome containing cells before and after differentiation revealed an induction of involucrin protein expression. However, there was a significant attenuation of E1^E4 protein expression when CTCF binding to the viral genome was abrogated (**Fig 7B and 7C**) and this was consistent in an independent keratinocyte donor (**S6 Fig**). Since L1 protein is not robustly expressed in methylcellulose differentiated keratinocytes, we analysed L1 protein expression by immunostaining organotypic raft culture sections derived from two independent donors of HPV18 and ΔCTCF-HPV18 genome containing cells. L1 positive cells were visible in the upper layers of HPV18 genome containing rafts but were barely detectable in ΔCTCF-HPV18 rafts and this difference was significant (**Fig 7D and 7E**). While the total number of E1^E4 positive cells in the upper layers of ΔCTCF-HPV18 rafts was not altered, the intensity of E1^E4 staining was notably reduced (**Fig 7D**). Western blot analysis of protein lysates harvested from three independent raft cultures confirmed a significant reduction in E1^E4 protein abundance in HPV18-ΔCTCF genome containing raft cultures in comparison to HPV18 (**Fig 7F**). Conversely, an increase in both E6 and E7 protein expression in raft lysates was observed (**Fig 7F**) while expression of E2 protein was not altered (**Fig 7G**), as previously reported [7] and in agreement with our Illumina and Nanopore RNA-seq datasets.

We previously demonstrated that in undifferentiated cells, ΔCTCF-HPV18 episomes had a higher abundance of trimethylation of lysine 4 in histone 3 (H3K4Me3) at the $P_{102}$ early promoter compared to HPV18, correlating with increased promoter activity and early transcript abundance. Interestingly, while differentiation of HPV18 genome-containing cells resulted in a significant enrichment of H3K4Me3 at the $P_{811}$ late promoter, no further enrichment above that observed in undifferentiated cells was observed in ΔCTCF-HPV18 episomes [6]. These data suggested that abrogation of CTCF binding resulted in an alternative epigenetic chromatin state of HPV18 episomes, driving enhanced early transcript production. However, we did not go any further to determine the impact of this altered chromatin state on late promoter activation and late gene transcription. To further understand the epigenetic changes that regulate promoter usage throughout the HPV18 life cycle, we opted to study the acetylation status of histone 4 (H4Ac), which is deposited downstream of H3K4Me3 and a hallmark of enhanced activation of transcription by facilitating increased chromatin accessibility and the recruitment of transcriptional activators [44]. H4Ac abundance in the viral genome in undifferentiated

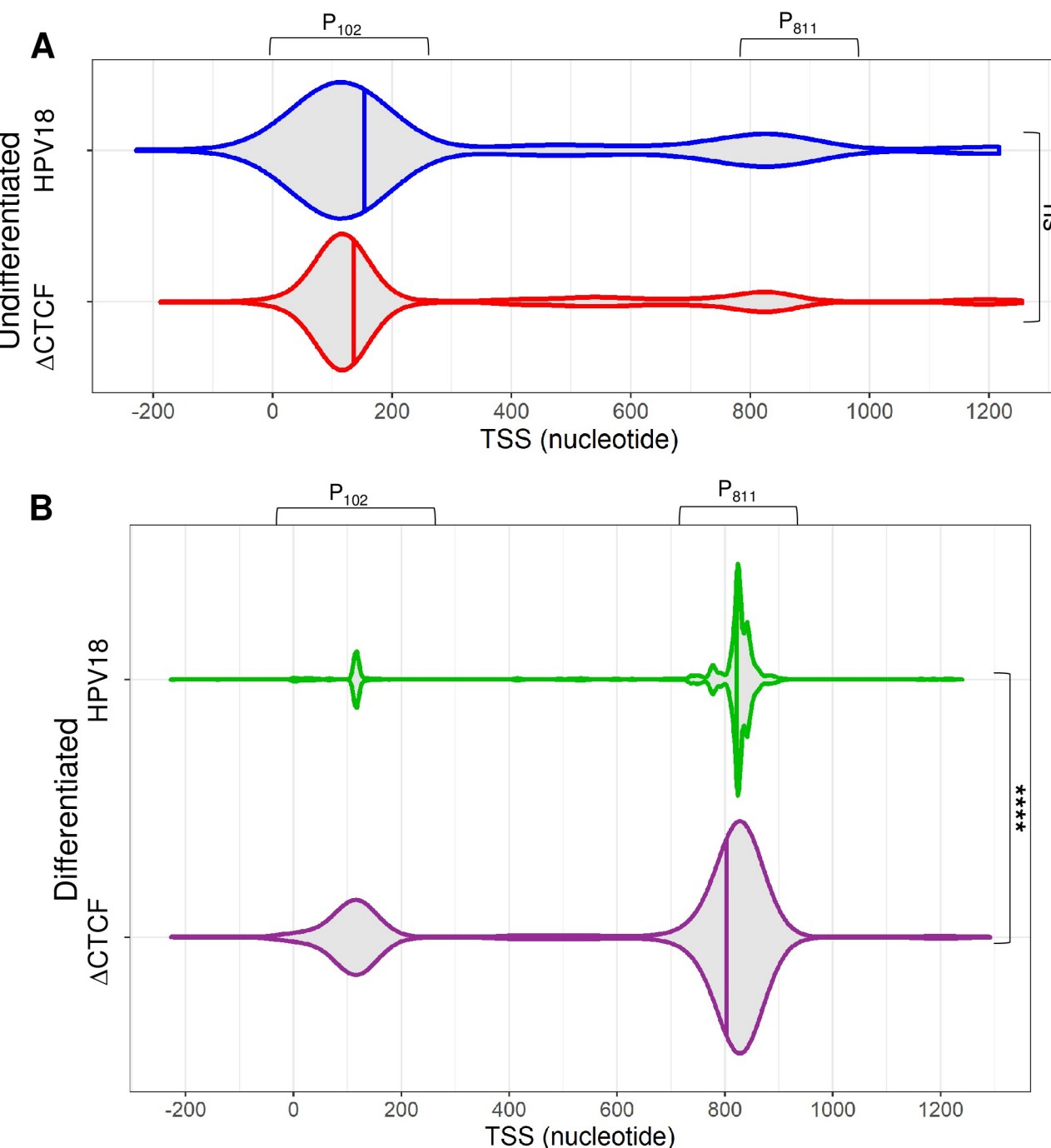

**Fig 6. CTCF regulates efficient differentiation-dependent HPV18 late promoter activation.** The 5' end of each viral transcript was identified and the distribution shown in violin plots in (A) undifferentiated and (B) differentiated keratinocytes containing HPV18 (blue and green, respectively) and ΔCTCF-HPV18 (red and purple, respectively) episomes. Data distribution are shown by the kernel shape and median indicated with a vertical solid line. The widest sections of each violin plot indicates the highest probability of promoter usage within that region of the HPV18 genome. The shape of the distribution indicates the concentration of data points in a particular region; the steeper the side of each bubble indicates a greater concentration of data points. ns, non significant; ****p<0.0001 (Fisher's test).

cells was detectable at low levels, consistent with restricted virus transcription (**Fig 8**). Differentiation of the cells in methylcellulose resulted in a dramatic increase in H4Ac abundance throughout the HPV18 genome, with an over 10-fold enrichment upstream of the $P_{811}$ late

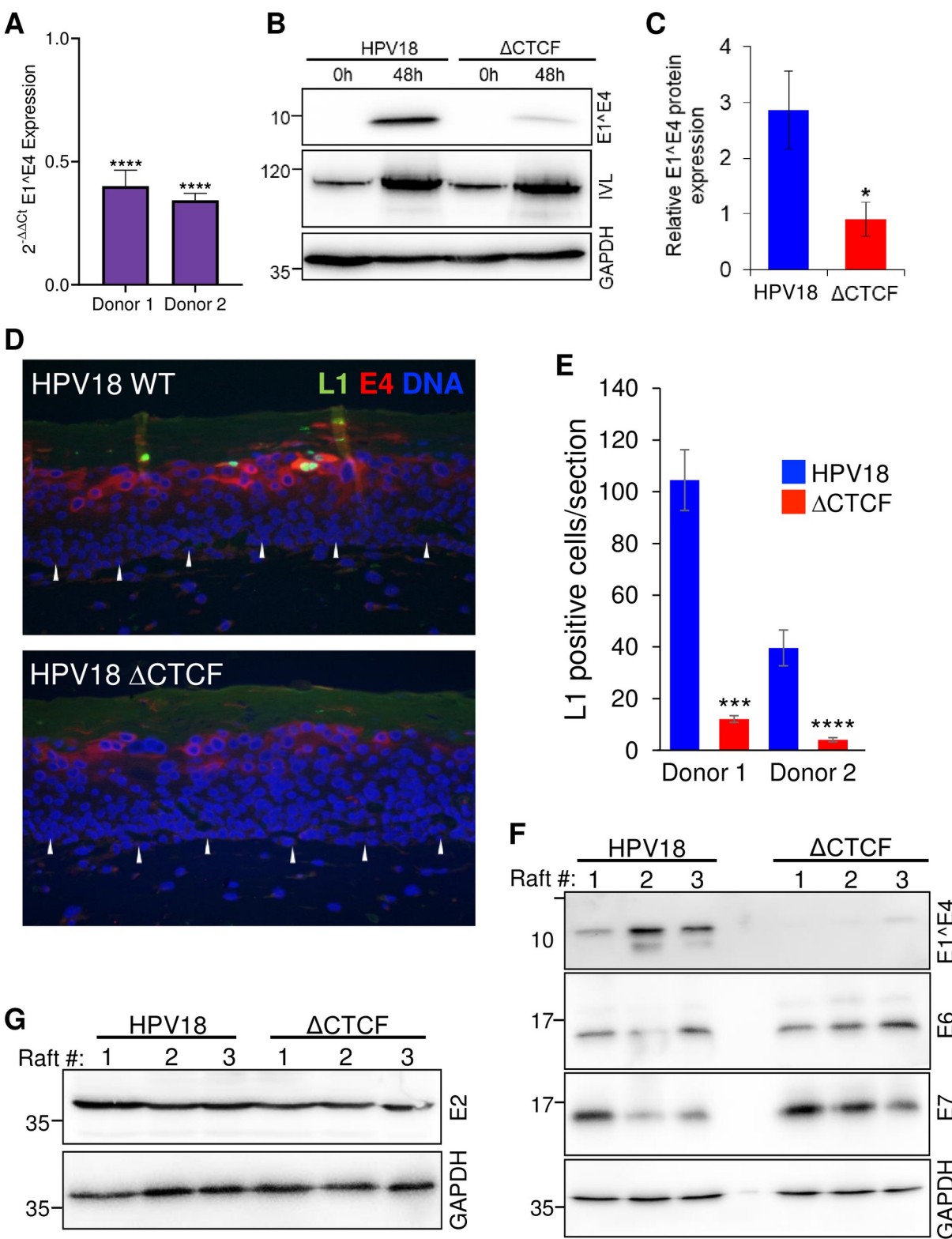

**Fig 7. Abrogation of CTCF binding to the HPV18 genome causes a significant reduction in differentiation-dependent late protein abundance.** (A) HPV18 genome containing keratinocytes (HPV18 or ΔCTCF-HPV18) grown in monolayer (undifferentiated, 0h) or differentiated in methylcellulose (48h) and E1^E4, involucrin (IVL) and GAPDH protein expression analysed by Western blotting. (B) Relative E1^E4 protein expression in comparison to GAPDH was quantified in three independent experiments by densitometry. Data are the

mean +/- standard deviation. * denotes p<0.05. (C) E1^E4 (red) and L1 (green) protein abundance was analysed by indirect immunofluorescence in epithelia derived from HPV18 and ΔCTCF-HPV18 genome-containing keratinocytes grown in organotypic raft culture. Cellular nuclei are shown in blue, and the basal layer indicated with white arrows. Scale bar indicates 10 μm. (D) The total number of L1 positive cells per section of three independent raft cultures grown from two independent keratinocyte donors was counted. Data show the mean +/- standard deviation. *** p<0.001, **** p<0.0001. (E) E1^E4, E6 and E7, and (F) E2 protein expression in organotypic raft cultures was assessed by Western blotting lysates harvested from three independent raft cultures alongside GADPH loading control. Molecular weight markers are indicated on the left of Western blots (kDa).

promoter, consistent with increased production of late transcripts. In contrast, H4Ac marks were barely detectable in ΔCTCF-HPV18 episomes in undifferentiated cells and only a small increase at the $P_{811}$ following differentiation was observed (Fig 8). However, it is important to note that H4Ac abundance at the $P_{811}$ promoter of ΔCTCF-HPV18 episomes was above that observed in undifferentiated HPV18 episomes, indicating attenuation rather than complete loss of activation of the HPV late promoter. These findings correlate with reduced late transcript abundance in differentiated ΔCTCF-HPV18 episomes compared to wild type. Together, these findings suggest that CTCF recruitment to the E2-ORF is necessary for appropriate epigenetic programming of the viral chromatin and differentiation-dependent transcriptional activation of $P_{811}$.

## Discussion

The differentiation-dependent regulation of papillomavirus transcription is fundamental to the productivity and persistence of infection. Previous studies have shown that the viral early

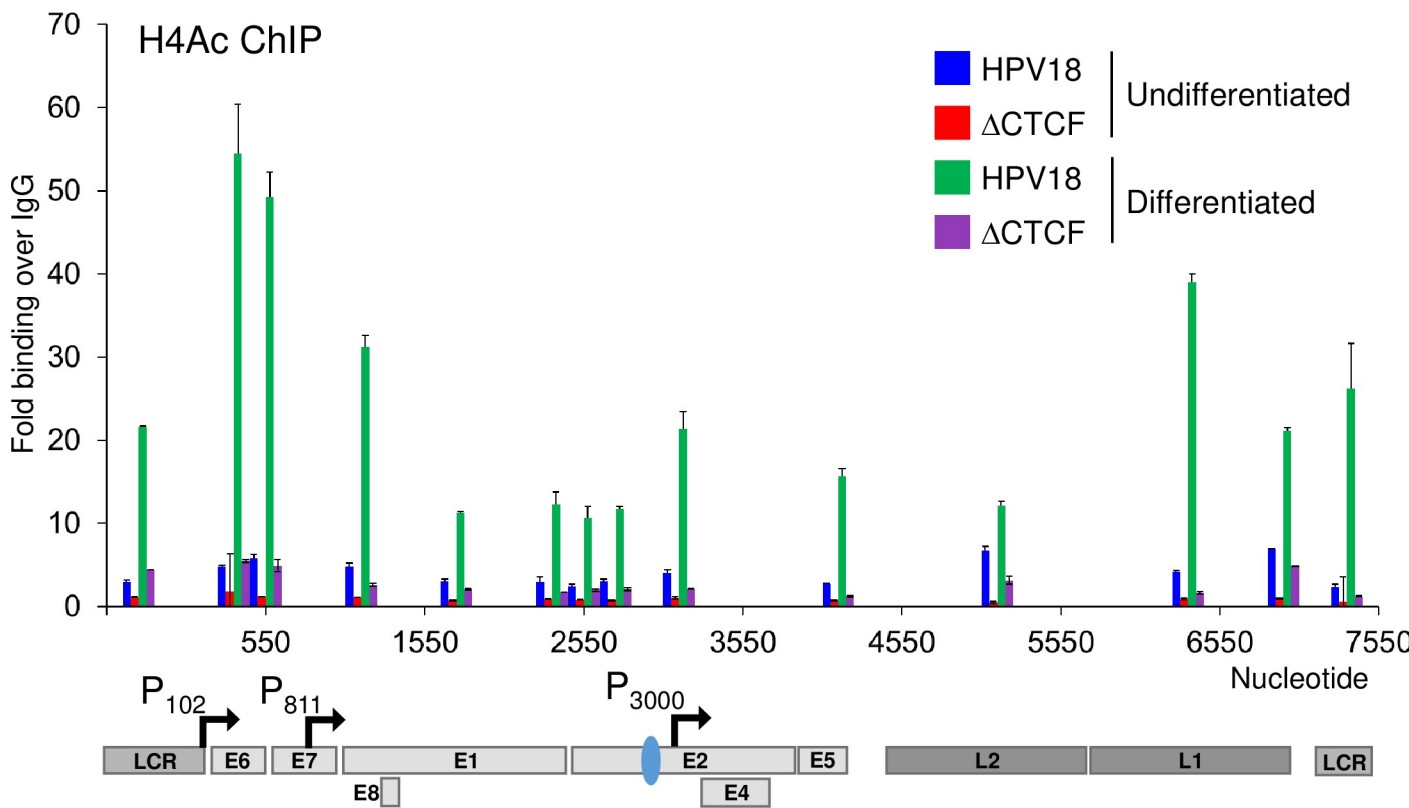

**Fig 8. Keratinocyte differentiation induces increased H4Ac abundance at the HPV18 late promoter in wild type but not ΔCTCF-HPV18 genome-containing cells.** HPV18 and HPV18-ΔCTCF genome-containing primary keratinocytes grown in monolayer (undifferentiated; blue and green, respectively) or differentiated in methylcellulose-containing media for 48 hrs (green and purple, respectively). Enrichment of H4Ac was assessed by ChIP-qPCR. Each bar in the chart represents the mid-point for primer pairs used to amplify immunoprecipitated chromatin. Fold binding over IgG control was calculated. The data shown are the mean and standard deviation of three independent replicates. Annotation of the HPV18 LCR, promoters, ORFs and CTCF binding site (blue oval) is provided below.

($P_{102}$) promoter is active in basal keratinocytes and becomes further activated as the cells enter terminal differentiation [5, 6]. In contrast, the viral late promoter ($P_{811}$) is repressed in undifferentiated basal cells and strongly activated upon induction of cellular differentiation [4, 5, 10, 17, 45]. In this study, we have utilised direct, long-read RNA sequencing to quantitatively analyse HPV18 promoter activity and to dissect the role of CTCF in regulating viral transcription at key stages of the virus life cycle. Our findings confirm the differentiation-dependent model of HPV transcription control; transcripts that originate from the $P_{102}$ promoter are dominant in undifferentiated cells and further increased in abundance upon cellular differentiation. The abundance of transcripts originating from the $P_{811}$ late promoter is low in undifferentiated cells but is dramatically upregulated when cells are differentiated. Transcription originating from the $P_{520}$ and $P_{3000}$ promoter regions is also activated by cellular differentiation but overall, these promoters are far less active than either the $P_{102}$ or $P_{811}$ promoters. The $P_{E8}$ promoter is equally weak in both undifferentiated and differentiated cells with only two transcript species that originate from this promoter region. The most dominant transcript identified from the $P_{E8}$ promoter was spliced at 1357^3434 and encodes E8^E2 and E5. The second transcript, spliced at 1357^3465 to encode E5 only, was slightly increased in expression in differentiated cell cultures.

Transcripts that encode fusion products between the E2 and E4 ORFs (E2^E4) have been previously described [41]. These transcripts were reported to originate upstream of the E2 start codon at position 2816 in HPV18 and therefore encode a protein fusion between the N-terminus of E2 and the C-terminus of E4. E2^E4S encoding transcripts, spliced at 2853^3434, were not identified in any of our Nanopore or RNA-Seq datasets. We did however detect transcripts spliced at 3165^3434, which have been previously described to encode a fusion protein termed E2^E4L [41]. However, this transcript was detected at very low abundance (~1 RPM) and only in differentiated keratinocytes. Interestingly, most of the transcripts that originated from the $P_{3000}$ promoter were also spliced at 3165^3434 or 3284^3434 (transcripts 8 and 9). These transcripts were in higher abundance than those originating from the $P_{102}$ promoter in both undifferentiated and differentiated cells, but since they lack the E2 start codon, they are likely to encode E5 protein only. Supporting this hypothesis, splicing of transcripts originating from the $P_{3000}$ promoter at 3165^3434 and 3284^3434 removes several intronic ATG start codons (7 and 11, respectively), potentially facilitating enhanced translation of E5.

Comparison of the HPV18 transcript map between HPV18 and ΔCTCF-HPV18 genome-containing cells revealed several important phenotypes. Firstly, abrogation of CTCF binding resulted in enhanced production of transcripts originating from the $P_{102}$ promoter, in agreement with our previous findings [6, 7]. The increased $P_{102}$ activity resulted in an increase in transcripts spliced at 233^416–929^3434 (encoding E6*I, E7, E1^E4 and E5) and 929^3434, (encoding E6, E7, E1^E4 and E5) while there was a small decrease in transcripts spliced solely at 233^416 (encoding E6*I, E1, E7 and E2) and 233^3434 (the only known transcript to encode E6*II), confirming our previous observation that abrogation of CTCF binding to the HPV18 genome reduces the abundance of transcripts spliced at 233^3434 [7]. In addition, a marked decrease in transcripts spliced at 3165^3434 and 3284^3434 was observed in ΔCTCF-HPV18 genome containing cells in comparison to HPV18, confirming our initial analysis of HPV18 transcript splicing by conventional RNA-Seq and validated by qRT-PCR in two independent keratinocyte donors. These data indicate that CTCF plays a key role in splice donor choice when splicing to the dominant splice acceptor site at nucleotide 3434 in the HPV18 genome.

A functional role for CTCF in influencing cellular co-transcriptional alternative splicing has been previously demonstrated. CTCF binding within or downstream of weak exons can promote exon inclusion by creating a roadblock to pause RNA polymerase II progression,

allowing greater splicing efficiency [23–25]. Interestingly, CTCF-mediated chromatin loop sta-bilisation between gene promoter and exon regions also plays a key role in regulating alterna-tive splicing events. Exons downstream of a CTCF stabilised promoter-exon loop are more likely to be included in the nascent mRNA, providing a functional link between three-dimen-sional chromatin organisation and splicing regulation [26]. Notably, we have previously shown that CTCF binding to the HPV18 E2 ORF stabilises a chromatin loop with the viral LCR [6]. This loop is positioned immediately upstream of weak slice donor sites at 3165 and 3284. Since CTCF binding loss results in decreased splicing at both 3165^3434 and 3284^3434 to produce E5 encoding transcripts, we hypothesise that CTCF chromatin loop formation plays an important role in HPV18 splice site choice. It also remains to be determined whether CTCF-directed splicing at the downstream SD sites is due to RNA polymerase II stalling via CTCF-mediated roadblock repression.

As expected, cellular differentiation strongly induced $P_{811}$ promoter activation in HPV18 episomes. However, ΔCTCF-HPV18 genome containing cells displayed a notable reduction in the abundance of transcripts originating from this promoter following differentiation. Differ-entiation dependent activation of the $P_{3000}$ promoter was also attenuated in HPV18 episomes unable to bind CTCF. In agreement with the observed differentiation induced activation of the $P_{811}$ and $P_{3000}$ promoters in HPV18 episomes, we demonstrated a marked increase in H4Ac enrichment, particularly in around the $P_{811}$ and $P_{3000}$ promoters. Interestingly, a similar level of H4Ac enrichment following differentiation was not recapitulated in ΔCTCF-HPV18 epi-somes, indicating that CTCF binding to the E2-ORF is important for enhanced transcriptional activation in the late stages of the virus life cycle, either through direct mechanisms or indi-rectly via increased E6/E7 expression. Importantly, attenuation of differentiation-dependent late promoter activation in ΔCTCF-HPV18 resulted in significantly reduced E1^E4 protein expression following methylcellulose differentiation and a marked reduction in L1 protein expression in stratified epithelia. These results demonstrate for the first time that CTCF has essential functions in differentiation-dependent transcriptional dynamics in the productive phase of the HPV life cycle.

## Supporting information

**S1 Fig. Limited alteration of host CTCF binding by ΔCTCF-HPV18 episome establishment compared to HPV18.** (A) Venn diagram of CTCF peak regions in cells containing either HPV18 or ΔCTCF-HPV18 episomes showing the average total number of peaks present in two independent replicates within each condition as well as the number of overlapping and unique peaks. (B) Heatmap visualization of CTCF ChIP-Seq replicates from two independent HFK donors (#1 and #2) and corresponding input sample centered on the combined peak regions detected in HPV18 and/or ΔCTCF-HPV18 samples. (C) Scatter plots of pairwise sam-ple comparisons show high correlation between replicates as well as between HPV18 and ΔCTCF-HPV18 samples. Pearson's correlations coefficients (r) are given in the plots and are above 0.93 in any pairwise comparison.
(TIF)

**S2 Fig. Southern blot analysis of episome copy number and methylcellulose-induced genome amplification in HPV18 and ΔCTCF-HPV18 episome containing cells (donor 2).** Amplification of HPV18 and ΔCTCF-HPV18 episomes was detected by Southern blotting fol-lowing digestion with *Eco*RI to linearise the HPV18 episomes, or *Bgl*II which digests cellular DNA only (OC, open circle; L, linear; SC, supercoiled).
(TIF)

**S3 Fig. Global analysis of differentiation-dependent changes to host gene expression.** (A) PCA of host cell transcriptome in undifferentiated HFKs containing HPV18 (blue) and ΔCTCF-HPV18 (red) episomes and following 48hr incubation in methylcellulose (green and purple, respectively). Close clustering of HPV18 and ΔCTCF-HPV18 samples is observed in both undifferentiated and differentiated cell populations, indicating similar transcriptional profiles. Clear separation in PC1 is induced by host cell differentiation. (B-D) Gene expression changes in undifferentiated (red) and differentiated (purple) ΔCTCF-HPV18 genome-containing HFKs were analysed by long read Nanopore RNA-Seq, demonstrating enhanced involucrin (IVL) expression (B) and enhanced ECM1 expression combined with differentiation-induced exon 7 skipping in transcript variant 3; exon numbering and transcript variants are indicated to the right and below the ECM1 gene annotation (C). (D) Gene set enrichment analysis of differentiation-induced host differential gene expression in HPV18 and ΔCTCF-HPV18 episome containing HFKs. The top 10 most significant terms in Gene Ontology set; Biological Processes are shown with associated p value (-log10).
(TIF)

**S4 Fig. Differentiation-induced expression changes of genes associated with keratinocyte differentiation.** Heatmap showing differentiation-induced expression changes of genes within Biological Processes term GO:0030216:Keratinocyte Differentiation with a mean normalised count of >10 in HPV18 and ΔCTCF-HPV18 genome containing HFKs.
(TIF)

**S5 Fig. Chromosomal location of human-HPV18 fusion transcripts detected by nanopore sequencing of HPV18 and ΔCTCF-HPV18 episome containing HFKs.** Approximate location of human-HPV fusion transcripts is highlighted on the karyotype (image from BioRender) for HPV18 (blue) and ΔCTCF-HPV18 (red). Where multiple transcripts with identical virus-host fusion co-ordinates were identified, the number of reads is indicated.
(TIF)

**S6 Fig. Abrogation of CTCF binding to the HPV18 genome causes a reduction in differentiation-induced E1^E4 expression (donor 2).** HPV18 genome-containing keratinocytes (Donor 2; HPV18 or ΔCTCF) grown in monolayer (undifferentiated, 0h) or differentiated in methylcellulose (48h) and E1^E4, involucrin (IVL) and GAPDH protein expression analysed by Western blotting. Molecular weight markers are indicated on the left (kDa).
(TIF)

**S1 Table. Virus-host fusion transcripts identified by Nanopore analysis of HPV18 genome containing HFKs.** Showing the nearest annotated human gene, coordinates of identified transcripts mapped to the human (Hg19) and HPV genomes and the number of reads detected.
(DOCX)

**S2 Table. Virus-host fusion transcripts identified by Nanopore analysis of ΔCTCF-HPV18 genome containing HFKs.** Showing the nearest annotated human gene, coordinates of identified transcripts mapped to the human (Hg19) and HPV genomes and the number of reads detected.
(DOCX)

## Acknowledgments

We thank Dr. Joseph Spitzer and his patients for the collection and donation of foreskin tissue.

## Author Contributions

**Conceptualization:** Karen Campos-León, Ieisha Pentland, Andrew D. Beggs, Adam Grundhoff, Sally Roberts, Joanna L. Parish.

**Data curation:** Jack Ferguson, Karen Campos-León, Ieisha Pentland, Joanne D. Stockton, Thomas Günther.

**Formal analysis:** Jack Ferguson, Thomas Günther, Adam Grundhoff, Sally Roberts, Boris Noyvert, Joanna L. Parish.

**Funding acquisition:** Sally Roberts, Joanna L. Parish.

**Investigation:** Jack Ferguson, Boris Noyvert.

**Methodology:** Karen Campos-León, Ieisha Pentland, Joanne D. Stockton, Thomas Günther, Andrew D. Beggs, Adam Grundhoff, Boris Noyvert.

**Project administration:** Joanna L. Parish.

**Resources:** Andrew D. Beggs, Adam Grundhoff, Joanna L. Parish.

**Supervision:** Andrew D. Beggs, Adam Grundhoff, Sally Roberts, Joanna L. Parish.

**Validation:** Jack Ferguson, Boris Noyvert.

**Visualization:** Boris Noyvert, Joanna L. Parish.

**Writing – original draft:** Jack Ferguson, Joanna L. Parish.

**Writing – review & editing:** Jack Ferguson, Karen Campos-León, Thomas Günther, Sally Roberts, Boris Noyvert, Joanna L. Parish.

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
