## [Decision Letter · Decision Letter 0]

9 Jun 2021

Dear Dr. Parish,

Thank you very much for submitting your manuscript "The chromatin insulator CTCF regulates HPV18 transcript splicing and differentiation-dependent late gene expression" for consideration at PLOS Pathogens. As with all papers reviewed by the journal, your manuscript was reviewed by members of the editorial board and by several independent reviewers. In light of the reviews (below this email), we would like to invite the resubmission of a significantly-revised version that takes into account the reviewers' comments.

We cannot make any decision about publication until we have seen the revised manuscript and your response to the reviewers' comments. Your revised manuscript is also likely to be sent to reviewers for further evaluation.

Sincerely,

Paul Francis Lambert

Associate Editor

PLOS Pathogens

Karl Münger

Section Editor

PLOS Pathogens

Kasturi Haldar

Editor-in-Chief

PLOS Pathogens

orcid.org/0000-0001-5065-158X

Michael Malim

Editor-in-Chief

PLOS Pathogens

orcid.org/0000-0002-7699-2064

Reviewer's Responses to Questions

**Part I - Summary**

Reviewer #1: Ferguson et al. describe a long-read RNA-seq analysis of HPV transcripts using cells that maintain wt HPV 18 genomes in comparison to those with mutated CTCF sites in E2. Previous studies by this group showed that that loss of CTCF binding at this sE2 site results in increased early transcripts, altered splicing patterns and changes in chromatin factors. In the present study the authors use long-read RNA seq to extend the analysis to late transcripts upon differentiation. Mutation of the CTCF site reduces late E1^E4 transcripts by 50% with slightly greater decreases in protein levels. The most significant effect of mutating CTCF site is alteration in splice site utilization along with substantially reduced levels of H4Ac histone modifications.

This study provides useful information that extends the authors previous findings on altered splice site usage and epigenetic modifications of the HPV early region with the CTCF mutant genome. The manuscript could benefit from a more mechanistic analysis of how loss of CTCF binding leads to altered splice site choice and changes in histone modifications.

Reviewer #2: This manuscript hones in on the understanding of HPV early and late gene regulation. The group has previously identified that the genome-organising protein CTCF is involved in forming a chromatin loop that regulates HPV gene expression. The group uses Nanopore sequencing to understanding HPV early and late transcriptional regulation that appears to be regulated by the presence of a CTCF binding site in the E2 ORF. CTCF is emerging as a new player in regulating alternative splicing via co-transcriptional, genomic and epigenetic mechanisms. Even more recently CTCF has been shown to be involved in regulating splicing in different viral genomes. This is a timely study and uses long-read sequencing to quantify the contribution of CTCF in alternative splicing regulation and transcriptional control in HPV18. Figure 3 is particularly impressive and the authors should be commended. However, there are some deficiencies in results, data analysed and mechanistic understanding that need to be addressed or expanded.

Reviewer #3: Manuscript by Joanna Parish, et al. describes the chromatin insulator CTCF regulates HPV18 transcript splicing and differentiation-dependent late gene expression. The authors in this report first found a CTCF binding site in the N-terminal transacting active domain of E2 and then compared viral gene expression in HFKs transfected with recirculated wt HPV18 or �CTCF-HPV18 DNA initially by RNA-seq and then by Nanopore-seq (long reads sequencing) analyses using mRNAs extracted from the transfected HFKs. The authors found that the cells with �CTCF-HPV18 DNA transfection exhibited a much high level of viral early gene expression from P102 promoter by RNA-seq analysis (Fig. 1B). However, the data from Nanopore-seq did not show the similar profile with the enhanced expression of viral early genes in HFKs transfected with �CTCF-HPV18 DNA (Figs. 3-5). Although most viral splicing profiles and promoter usage were confirmative and the observation of CTCF binding to the E2 region to regulate viral genome expression is attractive, the data provided in this report are not conclusive, and more confirmation experiments are needed to draw a definite conclusion. The data in RNA-seq and Nanopore-seq appear to be compiled from only a single sample each in wt and �CTCF-HPV18 genome transfection. To avoid sample bias and outliers and batch effects in RNA-seq and Nanopore (long reads sequencing), minimal three RNA samples in high quality in each group are needed for statistical analysis. Like any other methods, the authors should also provide the perimeters or cutoff values used to count the real splice-junction reads and their long reads. Most importantly, introduction of point mutations to disrupt the CTCF binding site in the N-terminal transacting domain of E2 should not affect E2 protein translation and E2 function. Thus, functional E2 data from the �CTCF-HPV18 DNA-transfected HFKs should be provided to ensure that CTCF, not the E2, regulates viral promoter activities.

**Part II – Major Issues: Key Experiments Required for Acceptance**

Reviewer #1: 1). CTCF was shown to interact by looping with YY1 in the URR. Does mutation of the YYI site in the LCR recapitulate mutation of the CTCF site in E2 on late splicing effects? YY1 levels decrease upon differentiation so it is unclear how loss of CTCF looping affects late events. Could the effects on late splice site choice be mediated by altered levels of an early gene product that is affected by CTCF YY1 looping?

2). Is the 50% reduction in E1^E4 transcripts in the CTCF mutant the result of altered differentiation dependent replication or changes in cell cycle? Southern analysis should be performed to look at genome levels upon differentiation with the CTCF mutant genomes. Does the previously reported increase in E6/E7 transcription upon CTCF site mutation affect induction of late functions? Is reduced E1^E4 responsible for altered splicing patterns? Is there a difference in how CTCF affects splice site choice in the early region as opposed to late?

3). The reduction in H4Ac is interesting but the effects only reduce late promoter activity by 50%. Is there an increase in repressive histone marks or other modified histones on the late or early regions?

Reviewer #2: 1. Nearly one-third of the abstract is reporting previously published results. The focus should be more on the current study, the results and the implications/significance.

2. More details should be provided of the E2 ORF CTCF binding site. This should be given a precise location (other HPV genomic features are) rather than a generic location. A sequence and co-ordinates should be provided as well as an alignment with the CTCF core consensus motif.

3. ChIP sequencing nicely identified differences in binding to WT vs CTCFmutant HPV in infected keratinocytes. However, no changes in host CTCF genome-wide binding were mentioned. These should be analysed and described (at least in Supplementary data).

4. In Figure 1B what differences in host gene expression was observed? If there were changes in gene expression in HPV genome due to the deletion of the CTCF sites, then changes in host gene expression would also be expected. Please detail.

5. Examples of differentiation-dependent host cell gene expression in HPV-infected keratinocytes were provided (IVL and EMC1) but this was limited at best – cherry picking at worst. More genes need to be examined to create an overall picture. This could be in Supplementary.

6. Line 315: ‘Viral host fusion transcripts’ – Is there evidence, if any, that an absence of CTCF binding site in HPV genome could affect HPV viral integration? Loss of a CTCF site could affect episomal folding and/or association with the host genome and/or viral integration. Please comment in Discussion.

7. Why was H4 acetylation examined as opposed to other histone marks? This is not made clear.

8. The confirmation of differentially expressed HPV proteins by Western blot is a bit limited (Figure 7a & E) and only E1^E4 is shown. This reviewer would have more confidence in CTCF’s role in regulating late viral gene expression if more HPV proteins are examined. If possible, a viral protein that is predicted to remain unchanged (based on Nanopore sequencing) should also be included. E6 and E7 should be examined.

9. Figure 7A: The authors had previously shown CTCF expression increase upon HPV transduction. However, this has not been yet shown for differentiation. For completeness, total CTCF protein levels should also be examined.

10. A mechanism by which CTCF regulates alternative splicing is via the ‘roadblock model’ by pausing RNAPII elongation. Is RNAPII enriched at the HPV CTCF site during keratinocyte differentiation?

11. Are cohesin subunits eg RAD21 enriched at the HPV CTCF site and enforcing the CTCF-mediated chromatin loop?

12. By the authors own review (Ref 10), HPV episomal genome can be methylated. Is there any evidence that the E2 ORF CTCF binding site or other CTCF sites can be impacted by methylation. This is important as this could account for lack of CTCF binding during the HPV viral cycle. The authors should conduct COBRA or clonal bisulfite sequencing of the E2 ORF and other possible CTCF sites.

Reviewer #3: Fig. 1.

Lines 271-274 and Fig. 1A. Please show the details of the mutated sequences in the mapped CTCF binding site (nt 2960-3020) which is positioned at the N-terminal domain of E2. Hope the mutated sequences in the CTCF binding site are neutral (not being suppressive or enhancive to the viral genome expression!) and most importantly, will not inactivate the functional transactivation domain of E2, with normal E2 protein production.

Lines 280-285 and Fig. 1B. The increased splicing of 233^416 was most likely resulted from increased expression of delta CTCF genome because total viral reads were all increased across the entire virus genome! How did the authors confirm the same amount of recirculated wt and mt HPV18 DNA being transfected into the HFKs? Some quantitative verification data on input viral genome copies in transfected cells are needed at the beginning of the study.

Fig. 2C. How many repeats do you have? Is this just one sample data or one representative of three qualified Nanopore-seq? Plus, all RNA-seq raw data should be submitted to NCBI Geo before manuscript submission.

Fig. 3.

Lines 327-336 and Fig. 3 transcriptome summary. Additional methods should be applied to verify “novel transcripts” identified from this single sample study.

Lines 337-339. Not sure about E2^E4L from the reported study (reference #38) but should be verified too in this report.

Fig. 3. HPV18 transcripts of wt vs CTCF mt in HFKs under undifferentiated and differentiated conditions: As the reads are so low (many are less than 5 reads per million) in so called “new transcripts”, how did we know these splice-junction reads were not background noise reads. What were the bioinformatics perimeters and thresholds used in the analyses? Where are the verification data on these new transcripts? Obviously, the late transcripts are so low due to poor HFK differentiation under methylcellulose condition where the RNA was prepared for RNA-seq or Nanopore-seq.

Plus, all Nanopore-seq raw data should be submitted to NCBI Geo to obtain a Geo Accession number for manuscript inclusion and submission.

Fig. 4 and Fig. 5. Because the RNA 5’ cap prevents linker ligation, RNA-seq and Nanopore-seq can’t be used to precisely map a transcription start site. What the authors’ reads counts in the two figures are the reads counts downstream of a real TSS.

Fig. 4B. Since only one or two reads were identified at each nt position in the virus genome, it would be better to delete those reads or state clearly in text that the biological functions of these rare TSSs are minimal.

Fig. 4D. If possible, please verify the promoter P3000 TSS by 5’ RACE.

Line 410-411 and Fig. 5. Need to be careful on the statement only until the authors ensure a functional E2 expression from �CTCF-HPV18 virus and the HFKs receiving the same amount of wt and mt HPV18 DNA. What are reads-scales in the Fig. 5A and 5B? To this reviewer, both the P102 and the P811 promoters appear more active from the �CTCF-HPV18 virus than wt HPV18 virus in differentiated HFKs, despite that the authors claim only the P102, but not the P811, being more active in the differentiated HFKs. The difference in Fisher’s test must be showing more transactivation in �CTCF-HPV18 virus than wt HPV18.

Fig. 6 and Fig. 7.

Fig. 6. Please make the annotated genome positions in scale to relative H4Ac positions. Where is your Fig. 6B (lines 24-426)?

Line 429. Not sure this statement is correct on reduction of P811 promoter activation from �CTCF-HPV18 virus in differentiated HFKs (Fig. 5B).

Fig. 7D and 7E. Now we can see the sample variations when three biological samples were examined. Thus, the same principle should be applied to all RNA-seq and Nanopore-seq for the data shown in Figs. 1-5.

**Part III – Minor Issues: Editorial and Data Presentation Modifications**

Reviewer #1: (No Response)

Reviewer #2: 1. Line 134: splicing

2. Include antibody dilutions or amounts used in each application

3. Include amplicon size in Table 1

4. It is not obvious how images (Figure 7) were scored or analysed. Was it done by image analysis or by a researcher blinded to the experimental arm?

5. A scale and 0- Z(?) x-co-ordinates should be provided on each HPV plot. At a suggestion, the CTCF site should be indicated with a black triangle or similar on Figure 1B & 6.

6. Figure 1 ‘wild type’ is used, but HPV18 is used thereafter. Please be consistent with usage in all Figures and text.

7. Figure 1 Legend: ‘Next generation sequencing data were IGV.’ The sentence isn’t complete?

8. Figure 2B – number and label the exons. No mention of EMC1 isoform(?) numbers 1, 2 & 3 are mentioned in the legend.

9. Lines 513-514 – ‘….in virus unable to bind CTCF’. Please rectify.

Reviewer #3: Introduction

Lines 97-103. In addition to CTCF, YY1 also interacts with HPV E7.

Line 105. It would be better to describe this promoter as P102 instead of P105 for HPV 18 E6/E7 and make it consistent in the entire manuscript. In fact, the main TSS in raft cultures starts from the P102, not much so from the P105.

Lines 119-123. The full HPV18 transcriptome has been summarized in a JVI article (JVI 90: 9138-952, 2016) and host SRSF3 and hnRNPA1 regulation on HPV18 RNA splicing has been described in the same paper.

Discussion

Please make the promoter P102 as a name consistently in the entire manuscript, including figures and text.

Lines 469-483. These spliced transcripts are rare and should not be emphasized too much. In your best estimate, they are less than 10 reads per million both in undifferentiated and differentiated conditions (Fig. 3)!

Lines 484-509. Because the RNA-seq and Nanopore-seq lack qualified biological sample repeats and there was no data on functional E2 data from �CTCF-HPV18 virus, this paragraph is not relevant.

PLOS authors have the option to publish the peer review history of their article (what does this mean?). If published, this will include your full peer review and any attached files.

Reviewer #1: No

Reviewer #2: **Yes: **Dr Chuck Bailey

Reviewer #3: No
---

## [Editor Report · Decision Letter 1]

13 Oct 2021

Dear Professor Parish,

We are pleased to inform you that your manuscript 'The chromatin insulator CTCF regulates HPV18 transcript splicing and differentiation-dependent late gene expression' has been provisionally accepted for publication in PLOS Pathogens.

Best regards,

Paul Francis Lambert

Associate Editor

PLOS Pathogens

Karl Münger

Section Editor

PLOS Pathogens

Kasturi Haldar

Editor-in-Chief

PLOS Pathogens

orcid.org/0000-0001-5065-158X

Michael Malim

Editor-in-Chief

PLOS Pathogens

orcid.org/0000-0002-7699-2064
---

## [Editor Report · Acceptance letter]

29 Oct 2021

Dear Professor Parish,

We are delighted to inform you that your manuscript, "The chromatin insulator CTCF regulates HPV18 transcript splicing and differentiation-dependent late gene expression," has been formally accepted for publication in PLOS Pathogens.

Best regards,

Kasturi Haldar

Editor-in-Chief

PLOS Pathogens

orcid.org/0000-0001-5065-158X

Michael Malim

Editor-in-Chief

PLOS Pathogens

orcid.org/0000-0002-7699-2064